# DNA-PAINT MINFLUX nanoscopy

Lynn M. Ostersehlt [1,6], Daniel C. Jans [1,2,6], Anna Wittek [1,2], Jan Keller-Findeisen[1],
Kaushik Inamdar [1,2], Steffen J. Sahl [1], Stefan W. Hell [1,3,4] ✉ and Stefan Jakobs [1,2,4,5] ✉

**MINimal fluorescence photon FLUXes (MINFLUX) nanoscopy, providing photon-efficient fluorophore localizations, has brought about three-dimensional resolution at nanometer scales. However, by using an intrinsic on–off switching process for single fluorophore separation, initial MINFLUX implementations have been limited to two color channels. Here we show that MINFLUX can be effectively combined with sequentially multiplexed DNA-based labeling (DNA-PAINT), expanding MINFLUX nanoscopy to multiple molecular targets. Our method is exemplified with three-color recordings of mitochondria in human cells.**

The MINimal fluorescence photon FLUXes (MINFLUX) imaging concept separates individual fluorophores at subdiffraction distances by switching them randomly 'on' and 'off', while establishing their position with an excitation light pattern featuring one or more intensity zeros, such as a donut[1,2]. Probing the fluorophore position with the central zero of a donut-shaped excitation beam substantially increases the localization precision for a given number of detected fluorescence photons. Previous studies showed that around 2,500 photons suffice to obtain precisions <1 nm (standard deviation) in the focal plane. Likewise, roughly 2 nm precisions were attained in three dimensions, demonstrating the capability of MINFLUX nanoscopy to resolve the spatial distribution of fluorophores at molecular scales[3,4].

All fluorescence nanoscopy concepts distinguish neighboring fluorophores by consecutively transferring them from a dark 'off' to a detectable 'on' state and back[5]. To this end, MINFLUX nanoscopy has so far relied exclusively on switchable or activatable fluorophores, that is molecules where the 'on/off' switching is afforded by state transitions within the fluorophore itself. However, the use of intrinsic state transitions places several constraints on the fluorophore, including on the brightness and the switching kinetics[2–4,6–11]. This is especially disadvantageous in multicolor recordings since the brightness and switching kinetics of different fluorophores have to be matched within a narrow range, often by applying specific buffers. Initial MINFLUX implementations were limited to two-color recordings.

In DNA-based point accumulation for imaging in nanoscale topography (DNA-PAINT) nanoscopy the 'on/off' modulation is implemented differently, namely by transient binding of diffusing fluorescently labeled oligonucleotides (denoted as imager strands) to complementary docking strands that are conjugated to a target protein such as an antibody[12–14]. While diffusing fluorophores contribute less detectable fluorescence and hence are largely 'off', bound fluorophores are 'on' because they deliver a stream of fluorescence photons from a fixed coordinate until bleaching or dissociation of the imager strand (which equals to going back 'off'). Since the

fluorophore does not need to be intrinsically switchable or activatable, bright and stable fluorophores can be employed.

As it typically uses widefield illumination and recording with a camera, establishing the 'on/off' state contrast in DNA-PAINT is challenged by the 'background' fluorescence from diffusing ('off' state) imager strands. This is particularly true when the desire to increase imaging speed calls for high concentrations of diffusing imager strands. As a result, most cellular DNA-PAINT recordings are performed in the total internal reflection fluorescence or highly inclined and laminated optical mode[15,16].

We reasoned that by combining DNA-PAINT with MINFLUX recording, we could synergistically benefit from the advantages of both methods. As in the current MINFLUX nanoscopy implementations, the 'background' fluorescence stemming from diffusing imager strands is suppressed by the confocal pinhole, DNA-PAINT MINFLUX nanoscopy can be used in the far-field mode. DNA-PAINT MINFLUX nanoscopy is expected to provide the same single-digit nanometer resolution as conventional MINFLUX nanoscopy. As the state-switching kinetics are determined by the binding of an imager strand to a docking strand, no dedicated buffer systems are required and the kinetics can be adapted to the density of the targets by tuning the concentration of the imager strand. As in conventional MINFLUX nanoscopy using photoswitchable dyes, also in DNA-PAINT MINFLUX nanoscopy the individual localizations are recorded one by one. Thus, the imaging time scales with the number of targets, making single-beam scanning MINFLUX particularly suited for recording small regions of interest. An intrinsic benefit of using PAINT is the fact that when densely packed molecules are imaged, successive fluorophore docking avoids the interaction of fluorophores belonging to neighboring target molecules. Hence coactivation and mutual fluorophore quenching is largely avoided. Finally, as multiple orthogonal imager strands can be applied sequentially, each binding to a different docking strand (Exchange DNA-PAINT)[17], addressing multiple targets should also be straightforward. For an overview of synergies, see Extended Data Fig. 1.

To establish DNA-PAINT MINFLUX nanoscopy, we first systematically explored the influence of the experimental key parameters laser power, confocal pinhole diameter and imager strand concentration on the image quality and the recording speed. Specifically, we determined the influence of these three parameters on (1) the time between valid events ($t_{btw}$), (2) the background emission frequency ($f_{bg}$), which is determined by the microscope, (3) the center-frequency ratio (CFR), a filter parameter for localizations during image acquisition[4] and (4) the localization precision ($\sigma_r$). These parameters together provide a measure of the image quality, the average success of the localization process and the time for

[1]Department of NanoBiophotonics, Max Planck Institute for Multidisciplinary Sciences, Göttingen, Germany. [2]Department of Neurology, University Medical Center Göttingen, Göttingen, Germany. [3]Department of Optical Nanoscopy, Max Planck Institute for Medical Research, Heidelberg, Germany. [4]Cluster of Excellence 'Multiscale Bioimaging: from Molecular Machines to Networks of Excitable Cells' (MBExC), University of Göttingen, Göttingen, Germany. [5]Translational Neuroinflammation and Automated Microscopy, Fraunhofer Institute for Translational Medicine and Pharmacology ITMP, Göttingen, Germany. [6]These authors contributed equally: Lynn M. Ostersehlt, Daniel C. Jans. ✉e-mail: shell@mpinat.mpg.de; sjakobs@gwdg.de

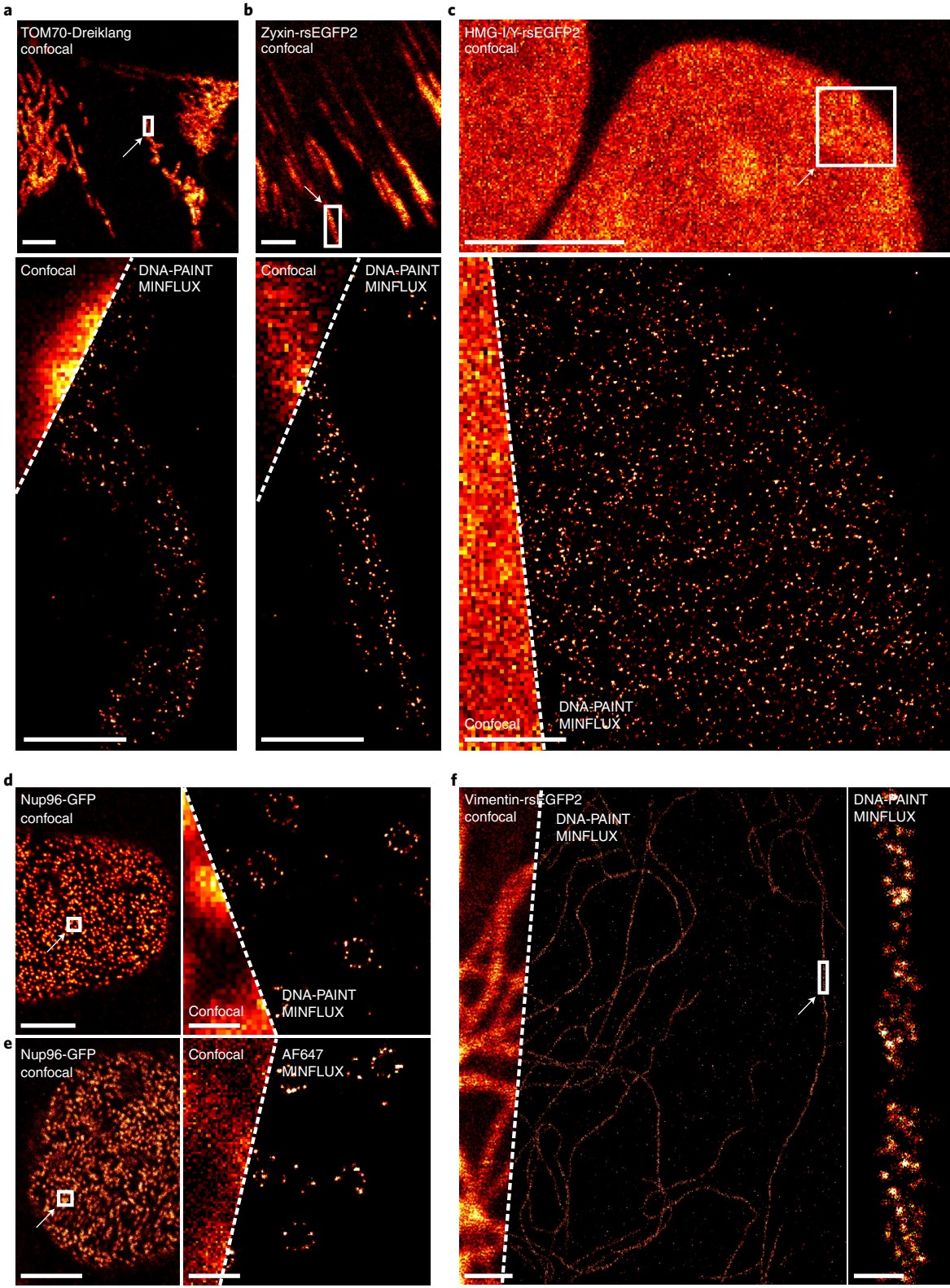

**Fig. 1 | 2D DNA-PAINT MINFLUX imaging. a–f,** Genome-edited cell lines expressing translational fusions with a fluorescent protein from the respective native genomic loci, as indicated (TOM70-Dreiklang (**a**), Zyxin-rsEGFP2 (**b**), HMG-I/Y-rsEGFP2 (**c**), Nup96-GFP (**d**, **e**), Vimentin-rsEGFP2 (**f**)), were labeled with an anti-GFP nanobody coupled to a docking strand and mounted with the imager strand (**a–d,f**), or with an anti-GFP nanobody coupled to Alexa Fluor 647 and mounted in STORM imaging buffer (**e**). Confocal overview images of the fluorescent protein fluorescence were taken. The rectangles indicate areas of MINFLUX recordings. For imager strand concentrations and localization precisions, see Supplementary Table 1. All Scale bars (confocal images): 5 μm (**a–e**), 1 μm (**f**). Scale bars (MINFLUX) 0.5 μm (**a–c**), 1 μm (**f**), 200 nm (**d,e**). Scale bar (MINFLUX close-up), 50 nm (**f**).

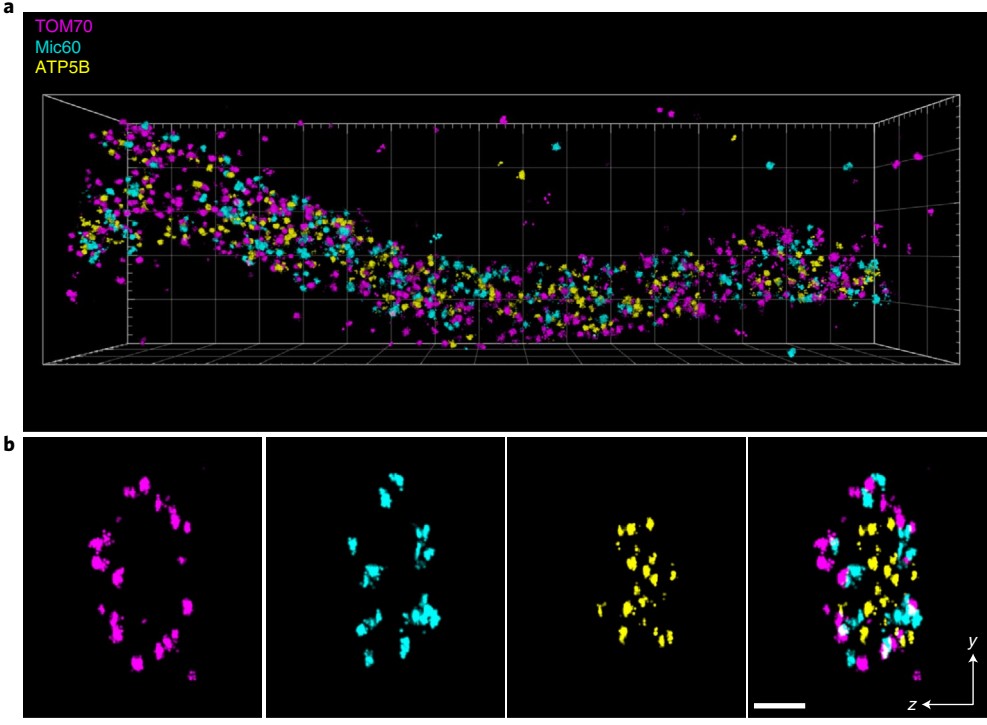

**Fig. 2 | 3D DNA-PAINT MINFLUX multiplexing.** U2OS TOM70-Dreiklang cells were fixed and immuno-labeled with an anti-GFP nanobody and anti-Mic60 and anti-ATP5B synthase antibodies. MINFLUX recordings of the three proteins were performed sequentially by adding and washing out the respective imager strands. Localizations of TOM70, Mic60 and ATP5B are displayed in magenta, cyan and yellow, respectively. **a**, View on a mitochondrial tubule. Size of the bounding box was $3.4 \times 1 \times 0.6\,\mu m^3$. **b**, Cross section of the tubule shown in **a**. Thickness of the section 100 nm. Scale bar 100 nm.

recording a DNA-PAINT MINFLUX image. For a description of the detailed analysis, see Supplementary Notes. In brief, we found that in DNA-PAINT MINFLUX imaging a sufficiently low imager strand concentration is a key determinant of the localization precision; a too low imager strand concentration, however, increases $t_{btw}$, and thereby the overall recording time. Reducing background by decreasing the pinhole diameter improves the localization precision. Conversely, a smaller pinhole increases $t_{btw}$, ultimately requiring the identification of an optimal pinhole size. The localization precision scales with increasing laser intensity. At the available laser intensities, we did not observe a relevant effect on $t_{btw}$. The CFR proved to be an easily accessible and reliable indicator for the expectable localization precision within an experimental series. For measurements with Atto 655 bound to the imager strand, the analysis suggested as a good starting point an excitation laser power (at 640 nm) of roughly 62 μW in the sample for the first MINFLUX iteration, a pinhole diameter of 0.4 Airy units (AU) and, for proteins with an overall density similar to nuclear pore proteins, an imager strand concentration of 2 nM.

Using these parameters, we first recorded two-dimensional (2D) DNA-PAINT MINFLUX images of various cellular structures exhibiting different densities of the target proteins (Fig. 1). To this end, chemically fixed genome-edited U2OS and HeLa cell lines, endogenously expressing fusions of a host protein and a fluorescent protein, were used. The fluorescent proteins were decorated with a nanobody featuring a docking strand. Confocal overview images recording the fluorescent protein fluorescence were used to identify smaller regions that were subsequently imaged by MINFLUX nanoscopy. As the cell lines expressing TOM70-Dreiklang (mitochondrial outer membrane), Nup96-GFP (subunit of the nuclear pore complex) and Vimentin-rsEGFP2 (vimentin cytoskeleton) exhibited a moderate target protein density, we chose an imager strand concentration of 2 nM. For the Zyxin-rsEGFP2 expressing cells that exhibited

a slightly less dense distribution of target proteins, we chose 2.5 nM imager strand. For all images, the localization precision ($\sigma_r$) of an individual localization event was in the range of 2.4 to 2.7 nm (Supplementary Table 1 and Extended Data Fig. 2); note that the images display all recorded valid localization events (Fig. 1). As previously demonstrated, the individual localizations of single binding events can also be combined[7], resulting in higher nominal localization precisions of 0.8 to 1.1 nm ($\sigma_{rc}$) (Supplementary Table 1). The localization precisions achieved with DNA-PAINT MINFLUX were comparable to the localization precision achieved when using the photo-switching of Alexa Fluor 647 for MINFLUX nanoscopy ($\sigma_r = 3.0$ nm; $\sigma_{rc} = 1.4$ nm) (Fig. 1e and Supplementary Table 1).

To investigate whether DNA-PAINT MINFLUX nanoscopy is indeed suitable for addressing densely packed protein distributions, we imaged U2OS cells in which the abundant nonhistone chromatin protein HMG-I/Y was endogenously tagged with the fluorescent protein rsEGFP2. An imager strand concentration of 0.5 nM enabled recordings of the distribution of HMG-I/Y (Fig. 1c) with a localization precision ($\sigma_r$) of 2.3 nm (Supplementary Table 1 and Extended Data Fig. 2).

Cryo-electron tomography of in vitro reconstituted vimentin filaments suggested the assembly of four-stranded protofibrils with a right-handed supertwist[18]. In parts of the DNA-PAINT MINFLUX recorded filaments we indeed identified patterns that were highly suggestive of a twist, whereas in other parts this was not obvious (Fig. 1f). To determine whether insufficient sampling of localization events was causing these differences in the visibility of twists, we analyzed the accumulated localizations at different time points during a prolonged DNA-PAINT MINFLUX recording. Visual inspection suggested that in the first 4–5 hours of the DNA-PAINT MINFLUX recording new localizations continuously enhanced the vimentin imaging, while after 6–7 hours, new localizations did not add to the vimentin structure (Extended Data Fig. 3). This impression

was fully in line with a Fourier ring correlation (FRC) analysis[19] of the images recorded at the different time points. After 6–7 hours, the FRC resolution value reached a plateau (Extended Data Fig. 3). We conclude that most of the accessible binding sites had been captured, and that a prolongation of the recoding time would not have improved the recording further. We also note that the progression of the FRC resolution values could be used as an abort criterion to determine the endpoint of DNA-PAINT MINFLUX recordings.

So far, all MINFLUX recordings were restricted to at most two-color channels. To demonstrate three-channel DNA-PAINT MINFLUX nanoscopy, we decorated TOM70-Dreiklang expressing cells with an anti-GFP nanobody conjugated to a docking strand, whereas the inner membrane proteins Mic60 and the subunit beta of the $F_1F_O$-ATP synthase (ATP5B) were labeled using specific antibodies. The three-dimensional (3D) MINFLUX imaging of the three channels was performed sequentially by adding and washing out the respective imager strands. The obtained localizations were rendered in three dimensions and overlaid (Fig. 2). The resulting 3D localization precision for all three target proteins was roughly 5.4 nm ($\sigma_r$) and 3.1 nm ($\sigma_z$) (Supplementary Table 1). When the individual localizations of single binding events were combined[7], we achieved localization precisions of roughly 2.0 nm ($\sigma_{rc}$) and 0.8 nm ($\sigma_{zc}$) (Supplementary Table 1).

Altogether, this study quantified the influence of the laser power, pinhole size and imager strand concentration on the image quality, thereby narrowing down the parameter space for future DNA-PAINT MINFLUX applications (Supplementary Notes). Since fluorescence microscopes render nothing but the fluorophores in the sample, the concept of optical resolution can only be applied to the fluorophores. To be able to draw meaningful conclusions about the target molecules at the <5 nm scale, the size and mobility of the linker between the molecule and the fluorophore have to be taken into account. To fully harness the nanometer optical resolution potential of MINFLUX nanoscopy, these sample parameters deserve further attention and improvement. In addition to the size of the label, the completeness of the labeling and the fraction of fluorophores that can be successfully localized must also be taken into account. DNA-PAINT MINFLUX makes it possible to localize each binding site several times. Therefore, missing localizations due to premature bleaching of the fluorophore are avoided with this technique.

Compared to initial MINFLUX implementations that relied on the popular STORM dyes Alexa Fluor 647, CF660C and CF680 as switchable fluorophores, (multicolor) DNA-PAINT MINFLUX nanoscopy is simpler to apply, as it avoids the use of complex buffer systems and the need to adjust to different blinking kinetics of the fluorophores in use (for a detailed comparison of the concepts, see Extended Data Fig. 1). The MINFLUX localization process remains unchanged compared to previous implementations. Therefore, DNA-PAINT MINFLUX nanoscopy provides the same unbiased, high-precision localization demonstrated in previous studies[2,3]. Finally, we note that by multiplexing DNA-PAINT labeling through the application of multiple orthogonal strands, our study paves the way for 3D MINFLUX imaging with nanometer resolution within cells with $n > 3$ channels.

## Online content

Any methods, additional references, Nature Research reporting summaries, source data, extended data, supplementary information, acknowledgements, peer review information; details of

author contributions and competing interests; and statements of data and code availability are available at https://doi.org/10.1038/s41592-022-01577-1.

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

## Methods

**Cell lines.** The genome-edited U2OS cell lines *HMGA1*-rsEGFP2 (homozygous), *Zyxin*-rsEGFP2 (homozygous) and *Vimentin*-rsEGFP2 (heterozygous) were described in ref. [20]. The heterozygous *TOMM70A*-Dreiklang U2OS cell line was generated as described in ref. [20]. The homozygous *NUP96*-mEGFP cell line U2OS-CRISPR-*NUP96*-mEGFP clone no. 195 (300174)[21] and the *NUP107*-mEGFP cell line HK-2xZFN-mEGFP-Nup107 (300676)[22] were purchased from CLS GmbH (CLS Cell Lines Service GmbH).

**Cell culture.** U2OS cells were cultivated in McCoy's 5a medium (Thermo Fisher Scientific), supplemented with 100 U ml$^{-1}$ penicillin, 100 μg ml$^{-1}$ streptomycin, 1 mM Na-pyruvate and 10% (v/v) FBS (Invitrogen) at 37 °C, 5% CO$_2$. HeLa Kyoto cells (HK-2xZFN-mEGFP-Nup107) were cultivated in DMEM, high glucose, GlutaMAX Supplement, pyruvate (Thermo Fisher Scientific), supplemented with 100 U ml$^{-1}$ penicillin, 100 μg ml$^{-1}$ streptomycin and 10% (v/v) FBS (Invitrogen) at 37 °C, 5% CO$_2$.

**Sample preparation.** The cells were cultured for 1 day on cover slips (Marienfeld) or in eight-well chambered cover slips (ibidi) and fixed in prewarmed 8% formaldehyde in PBS for 10 min. Fixed cells were permeabilized with 0.5% (v/v) Triton X-100 in PBS for 5 min. NUP107-mEGFP cells were fixed in 2.4% formaldehyde in PBS for 30 min at room temperature and after fixation incubated with 0.1 M NH$_4$Cl in PBS for 5 min. Then, NUP107-mEGFP cells were permeabilized with 0.25% (v/v) Triton X-100. Afterward, all cells were blocked in antibody incubation buffer (Massive Photonics) for roughly 30 min. The cells were incubated for 1 h with the MASSIVE-TAG-Q anti-GFP single domain antibody (Massive Photonics) or with the FluoTag-Q anti-GFP single domain antibody (conjugated with Alexa Fluor 647) (NanoTag Biotechnologies) in antibody incubation buffer (Massive Photonics) at a dilution of 1:100. The cells were then washed three times with 1× washing buffer (Massive Photonics). For multiplexing, the cells were fixed, permeabilized and blocked as described above. Afterward, the cells were incubated for 1 h at room temperature with primary antibodies against Mic60 (Proteintech) at a concentration of 1.235 μg ml$^{-1}$ and ATP synthase subunit beta (Abcam) at a concentration of 5 μg ml$^{-1}$ in antibody incubation buffer (Massive Photonics). After three washing steps with PBS, the cells were incubated with polyclonal secondary antibodies coupled to DNA-PAINT docking sites, targeting mouse and rabbit IgGs (Massive Photonics) at a dilution of 1:400 each and with MASSIVE-TAG-Q anti-GFP single domain antibody (Massive Photonics) at a dilution of 1:100. The cells were then washed three times with 1× washing buffer (Massive Photonics).

**Sample mounting and imaging buffer.** For the stabilization of the samples during MINFLUX imaging, the samples were incubated with 100 μl of gold nanorod dispersion (A12-40-980-CTAB-DIH-1-25, Nanopartz Inc.) for 7 min, as described before[3,4]. To remove unbound nanorods, the samples were rinsed with PBS several times. For single-color DNA-PAINT imaging, aliquots (5 μM) of the DNA-PAINT imager strand conjugated to Atto 655 (Massive Photonics) were diluted in imaging buffer (Massive Photonics) (final concentrations indicated in Supplementary Table 1). Alternatively, for MINFLUX imaging of Alexa Fluor 647, standard STORM buffer containing 10 mM MEA (Sigma-Aldrich), 64 μg ml$^{-1}$ catalase from bovine liver (Sigma-Aldrich), 0.4 mg ml$^{-1}$ glucose oxidase from *Aspergillus niger* (Sigma-Aldrich), 50 mM Tris/HCl, 10 mM NaCl and 10% (w/v) glucose, pH 8.0 was used[23]. Cover slips were sealed with picodent twinsil (picodent) on cavity slides (Brand GmbH & CO KG). For multiplexing, eight-well chambered cover slips (ibidi) were used. After incubation with gold nanorod dispersion and washing as described above, aliquots (5 μM) of the DNA-PAINT imager strand (conjugated to Atto 655) (Massive Photonics) transiently binding to MASSIVE-TAG-Q anti-GFP single domain antibody in imaging buffer (final concentration 2 nM) (Massive Photonics) and added to the cells. After DNA-PAINT MINFLUX imaging, the cells were washed on the microscope stage five times with PBS and one time with imaging buffer (Massive Photonics). Subsequently, DNA-PAINT imager (conjugated to Atto 655) (Massive Photonics) transiently binding to the anti-rabbit IgG was diluted (final concentration 1 nM) and added. After recording of the second DNA-PAINT MINFLUX dataset this process was repeated and imager transiently binding to the anti-mouse IgG (final concentration 1 nM) was added.

**MINFLUX measurements.** The data were acquired on an Abberior MINFLUX microscope (Abberior Instruments)[4] using Inspector Software (v.16.3.1 1647M-devel-win64-MINFLUX, Abberior Instruments). For MINFLUX measurements, the Inspector MINFLUX sequence templates seqIIF (2D) and DefaultIIF3D (3D) provided and optimized by the manufacturer for samples with the dye Alexa Fluor 647 were used (MINFLUX sequences).

   Cells were identified and placed in the focus using the 488 nm confocal scan of the microscope. If necessary, the persistent binding–unbinding activity of imager strands was verified in the 642-nm confocal scan. Before starting a MINFLUX measurement, the stabilization system of the microscope was activated. Measurements were conducted with a stabilization precision of typically below 1 nm. A region of interest was selected in the confocal scan image and laser power and pinhole size were adjusted in the software (indicated pinhole sizes in AU refer

to the emission maximum of Atto 655 at 680 nm). For MINFLUX measurements of Alexa Fluor 647 (Fig. 1e) a laser power of 12 μW in the first iteration and a pinhole diameter of 0.6 AU were used. Finally, the MINFLUX measurement was started in the region of interest.

*Quantification measurement series.* In a measurement series (Supplementary Notes) one of the experimental parameters, namely laser power, pinhole size or imager concentration, was varied, while the other parameters were kept constant. Within one measurement series, we recorded 2D MINFLUX images of labeled nuclear pores close to the cover slip and kept the image size and the recording time (1 h) constant. All images were taken with the same MINFLUX iteration sequence. Multiple regions (1 × 1 μm) of the lower envelope of one nucleus were measured. Each region was imaged with a different experimental parameter. Each measurement series was repeated three times on different days with fresh samples.

**Daily alignment of the MINFLUX nanoscope.** The shape of the intensity pattern ('donut') for fluorescence excitation was evaluated using immobilized fluorescent beads (GATTA-BEAD R, Gattaquant GmbH) and if necessary optimized by changing the spatial light modulator parameters. Additionally, the position of the pinhole was adjusted so that the confocal detection matched the excitation volume. If during measurement series more than one pinhole size was used, all pinhole positions were determined before starting the measurement series. The pinhole position was then adjusted before each measurement.

**MINFLUX data analysis.** *Data export.* Each MINFLUX measurement was exported using Inspector Software (Abberior Instruments). The exported files contained a collection of recorded parameters for all valid localizations and also included discarded nonvalid localization attempts. Additional information of the measurement (laser power and so on) was stored manually. Both were imported in a custom analysis script written in MATLAB (R2018b) to calculate the following quantification parameters in an automated manner.

*Quantification.* For all calculations, only data of the last MINFLUX iterations (in two dimensions fourth, in three dimensions nineth, after one prelocalization iteration), which were also identified as valid (exported parameter VLD = 1), were used.

   The first quantification parameter to be calculated was the time that passed between the localization of two valid events, in short, the time between events or $t_{btw}$. An emitting molecule is usually localized by the microscope several times in direct succession by repeating the last two MINFLUX iterations. These successive localizations are assigned to the same event via the same trace ID (exported parameter TID). Moreover, for each individual localization the time at which its localization process started was saved (exported parameter TIM). This allowed the determination of the start and end time of each molecule binding event. Each event (TID) was terminated after a predefined number of nonvalid localization attempts. The time of the first final nonvalid localization attempt was defined as the end time of the molecule binding event. Finally, $t_{btw}$ was calculated as the time difference between two consecutive valid events by subtracting the end time of the first molecule from the start time of the second molecule. For each measurement, the median of the first 100 events was determined as $t_{btw}$.

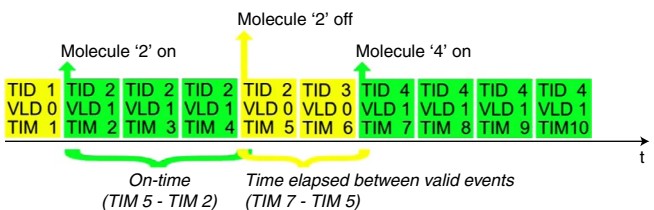

**Time between molecule binding events $t_{btw}$ calculated from the exported measurement parameters.** Saved localization attempts are depicted as colored rectangles, arranged in order of their appearance. Valid localization attempts were saved with the exported parameter VLD = 1 and are shown as green, while the nonvalid localization attempts were saved with VLD = 0 and are shown in yellow. The beginning of a localization attempt is saved as a time stamp (exported parameter TIM, here shown simplified as dimensionless values from 1 to 10. Localization events belonging to the same molecule have the same trace ID (exported parameter TID). Here, the time between the two consecutive valid molecules is calculated as the time difference between the start of molecule 4 (TIM = 7) and the end of molecule 2 (TIM = 5).

   The second quantification parameter was the background emission frequency ($f_{bg}$). The $f_{bg}$ is continuously estimated by the MINFLUX microscope between valid events and is used by the system to identify emission events and to correct emission frequencies of localization events.

   The third quantification parameter was the CFR. The CFR is the ratio of the effective emission frequency at the central position of the MINFLUX excitation pattern over the mean effective emission frequency over all outer positions and defined as CFR = $f_{eff}$(central position)/$f_{eff}$(outer positions). The effective

frequencies $f_{eff}$ are the measured emission frequencies above a background automatically determined by the system. The value of the CFR is regarded as a quality measure for the localization process. For each measurement, the median CFR of all valid localizations in the last iteration was determined. The CFR is calculated directly by the microscope software and is also used for filtering in early iterations (exported parameter CFR). It therefore directly influences the measurement[4].

To estimate the localization precision of a measurement as the third quantification parameter, the standard deviation $\sigma_r$ was calculated for each molecule (at least five localizations with the same exported parameter TID) as $\sigma_r = \sqrt{\left(\sigma_x^2 + \sigma_y^2\right)/2}$ with the standard deviations of the $x$ and $y$ coordinates as determined by the microscope (exported parameter POS). The median $\sigma_r$ represents the stated localization precision. The combined localization precision was estimated as $\sigma_{rc} = \langle\langle\sigma_r\rangle/\sqrt{n}\rangle_n$, that is the weighted average of the average single localization precision $\sigma_r$ divided by $\sqrt{n}$ and weighted by the occurrence of $n$ being the number of single localizations with the same TID. The precision in the z direction is often different from $x$ and $y$, therefore we separately computed the combined localization precision in z: $\sigma_{zc} = \langle\langle\sigma_z\rangle/\sqrt{n}\rangle_n$.

*CFR simulation.* The CFR is a parameter that is directly calculated during image acquisition by the MINFLUX software. To understand and judge the CFR values from the experimental results we simulated the CFR dependency on pinhole size and imager strand concentration for a molecule that is located at the center of the MINFLUX targeted coordinate pattern (TCP) with background contributions included (Supplementary Notes and Supplementary Fig. 3). The excitation PSF $h_{exc}(x, y, z)$ in shape of a 2D donut was determined via fast focus field calculations[24] for high numerical apertures and using realistic values for the objective lens properties as well as an excitation wavelength $\lambda_{exc} = 642$ nm. The confocal detection PSF $h_{det}(x,y,z)$ was calculated[25] for a detection wavelength of $\lambda_{exc} = 680$ nm. We then calculated the resulting effective PSF $h_{eff,i} = h_{exc,i}h_{det}$ for each exposure $i$ by shifting $h_{exc}$ to the according exposure position in the MINFLUX TCP while keeping the confocal detection $h_{det}$ centered. The background contribution due to diffusing imager strand was calculated in two steps. The resulting background intensity $B_i$ in the effective excitation volume was calculated as $B_i \approx \int_{x,y,z} h_{eff,i}(x, y, z) \times c_{imager} dxdydz$ for each exposure. For the CFR calculation, we assumed that the central donut exposure of the MINFLUX TCP is placed directly on the molecule, chosen here as the origin. In the case of a perfect donut zero, this leads to a detected emitter intensity of $I_{center} = 0$ for this exposure. The signal intensity detected at different exposures is calculated as $I_i \approx h_{eff,i}(0, 0, 0)$. Correcting for the different total time spent in the inner and outer exposures, the mean background intensity $\bar{B}_{outer}$ and mean signal intensity $\bar{I}_{outer}$ was calculated for the outer exposures ($i \neq 1$). Therefore, we were able to calculate the CFR as $\text{CFR} = \frac{B_{center} + I_{center}}{\langle \bar{B}_{outer} + \bar{I}_{outer} \rangle}$ for different scenarios. We repeated the calculations for different concentrations $c$, adapted the pinhole size when determining $h_{det}$ and used different values for the TCP diameter $L$.

*Sample drift correction.* Sample drift was corrected from the extracted molecule event position and time pairs by dividing the events into overlapping time windows of approximately 2,000 events per window, and generating a 2D or 3D rendered MINFLUX image (placing a Gaussian peak with standard deviation sigma of 2 nm at each estimated molecule position) and calculating 2D or 3D cross-correlations between images from different time windows. The center of the cross-correlation peak was fitted with a Gaussian function and its offset relative to the center of the cross-correlation presented the spatial sample shift between the corresponding time points. The drift curve that fulfilled all possible sample drift estimations for all possible time window pairs was estimated in a least squares sense. A smooth (cubic spline) interpolation of the estimated drift curve for all time points of all events was then subtracted from the molecule coordinates.

*FRC_{xy} calculations.* For the determination of the FRC shown in Supplementary Table 1 and Extended Data Fig. 3 we implemented the algorithm described in ref. [19]. In brief, a dataset of combined localizations (only $x$ and $y$ positions) was divided into two statistically independent subsets resulting in two subimages, each containing 50% of the combined localizations of the original dataset. Then, the average correlation of the Fourier transform of these subimages was calculated on rings of constant spatial frequency. The inverse of the spatial frequency at which the FRC drops below one-seventh was taken as a measure of the FRC resolution. We used combined localizations instead of single localizations for the estimation of the FRC resolution, because for single localizations the FRC is dominated by the large number of repeated localizations during one binding event and the calculated FRC resolution is then strictly proportional to the single localization precision. To obtain a more robust result, the random division into subsets was repeated several times and the obtained FRC resolutions for each division were averaged.

**Image rendering in two dimensions.** All valid localization events were rendered using Imspector Software and displayed as 2D histograms with the bin size 4 nm (Fig. 1a–f) and 1 nm (Fig. 1f, close-up).

**Image rendering in three dimensions.** Each MINFLUX measurement was exported with Imspector Software. The data were drift corrected (Sample drift correction) and the z position was scaled with the scaling factor 0.7 (ref. [3]). A rendering of the resulting localizations where each localization was replaced by a Gaussian peak with sigma 5 nm was imported into the Imaris Software (Imaris x64, v.9.7.2, Bitplane AG). The data were displayed as a blend volume rendition.

**MINFLUX sequences.** The MINFLUX microscope's data acquisition is controlled by a set of parameters that are specified within a text file (see seqIIF.json and seqDefaultIIF3d.json in the Supplementary Data). The set of parameters defines a sequence that controls the iterative zooming in on single molecule events and was provided and optimized by the manufacturer for samples with the dye Alexa Fluor 647. The MINFLUX iteration process is described in ref. [4]. In two dimensions, four iterations plus one prelocalization iteration were performed. In three dimensions, nine iterations plus one prelocalization iteration were performed. In the last iteration an $L$ of 40 nm was used. Key parameters of the 2D iteration sequence include:

| | TCP parameter $L$ (nm) | Photon limit (minimal photon count) | Dwell time (ms) | CFR limit | Laser power factor |
|---|---|---|---|---|---|
| Prelocalization | | 160 | ≥1 | Off | 1 |
| Iteration 1 | 288 | 150 | ≥1 | 0.5 | 1 |
| Iteration 2 | 151 | 100 | ≥1 | Off | 2 |
| Iteration 3 | 76 | 100 | ≥1 | 0.8 | 4 |
| Iteration 4 | 40 | 150 | ≥1 | Off | 6 |

**Supplementary software and data.** An additional software package is provided with the manuscript (https://doi.org/10.5281/zenodo.6563100) to facilitate reanalysis of the MINFLUX localization data. The package is written in MATLAB and contains experimental localization data of all recorded DNA-PAINT MINFLUX datasets, which are shown in this publication. The software applies analysis steps such as drift correction, precision estimation, as well as CFR and FRC calculations on each dataset.

**Availability of materials.** U2OS cells lines HMGA1-rsEGFP2, Zyxin-rsEGFP2, Vimentin-rsEGFP2 and TOMM70A-Dreiklang are available from the corresponding author upon reasonable request. All other materials are commercially available.

**Statistics and reproducibility.** All experiments in this paper were performed independently at least three times and yielded similar results.

**Reporting summary.** Further information on research design is available in the Nature Research Reporting Summary linked to this article.

## Data availability
All DNA-PAINT MINFLUX localization data have been deposited at https://doi.org/10.5281/zenodo.6563100. The raw data as provided by the microscope software are available at https://doi.org/10.5281/zenodo.6562764.

## Code availability
All custom codes used for image analysis are available at https://doi.org/10.5281/zenodo.6563100.

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

## Acknowledgements
We thank N. Molitor, E. Rothermel and T. Gilat for excellent technical assistance. We thank M. Ratz for creation of the genome-edited HMGA1-rsEGFP2, Zyxin-rsEGFP2, Vimentin-rsEGFP2 and TOMM70A-Dreiklang U2OS cell lines. We thank C. Brüser

for help with cell lines, imaging and discussions. We thank J. Pape for discussions and support. This work was supported by the Deutsche Forschungsgemeinschaft (DFG, German Research Foundation) under Germany's Excellence Strategy – EXC 2067/1-390729940 (to S.W.H. and S.J.) and by the European Research Council (ERC AdG no. 835102) (to S.J.). The MINFLUX system was funded by the DFG (grant no. INST 186/1303-1 to S.J.). The research was funded by the DFG-funded TRR 274 (project no. Z01) and SFB 1190 (project no. P01).

## Author contributions

L.M.O., D.C.J., S.W.H. and S.J. designed the research. S.W.H. and S.J. conceived the project. L.M.O., D.C.J., A.W. and K.I. performed research. L.M.O., D.C.J., J.K.-F., S.J.S., S.W.H. and S.J. analyzed data. D.C.J., S.J.S., S.W.H. and S.J. wrote the paper, with input from all authors. All authors approved the manuscript.

## Funding

## Competing interests

S.W.H. holds shares of Abberior Instruments and has revenues through MINFLUX patents held by the Max Planck Society. All other authors declare no competing interests.

## Additional information

**Extended data** are available for this paper at https://doi.org/10.1038/s41592-022-01577-1.

**Correspondence and requests for materials** should be addressed to Stefan W. Hell or Stefan Jakobs.

| | DNA-PAINT (with widefield localization) | DNA-PAINT MINFLUX | MINFLUX (with photoswitchable dyes e.g. Alexa Fluor 647) |
|---|---|---|---|
| **state-switching mechanism** | | ← same as DNA-PAINT | |
| **localization concept** | TIRF or HILO excitation widefield detection | → same as MINFLUX | MINFLUX excitation seq. confocal detection |
| **performance parameter** | | | |
| localization precision 2D | typically below 10 nm in biological samples. | same as MINFLUX | typically ~1 nm or below in biological samples. |
| localization precision 3D | lateral typically below 10 nm and axial typically below 50 nm in biological samples | same as MINFLUX (in this manuscript below 2 nm) | lateral and axial typically below 5 nm in biological samples. |
| state-switching kinetics | controlled by imager concentration, DNA sequence and buffer conditions | same as DNA-PAINT | controlled by light and buffer conditions |
| recording time | typically minutes - hours for up to ~80 x 80 µm$^2$ FOV | same as MINFLUX | typically minutes - hours for ~ 1 x 1 µm$^2$ FOV |
| number of channels/ multiplexing capability | theoretically unlimited due to multiplexing capability (Exchange-PAINT) | same as DNA-PAINT | max. 2 channels (by spectral separation of emitted photons of two photoswitchable dyes with similar excitation spectra) |
| applicability to high target density | easily adaptable to any target density (number of bound imager strands directly proportinal to imager concentration) | same as DNA-PAINT | limited adaptability via buffer composition (high density target imaging not possible). Mutual quenching or coactivation of dye molecules at high densities sometimes observed. |
| sample background | high (due to free imager in the buffer) | same as DNA-PAINT | low |
| method for background reduction | TIRF or HILO illumination | same as MINFLUX | confocal pinhole |
| imaging depth | <200 nm (TIRF); <10 µm (HILO) | same as MINFLUX | depth-imaging with abberation correction to be developed |
| photobleaching | no permanent photobleaching due to reservoir of imager in the buffer. Photobleaching during a single binding event possible. Binding site depletion possible. | same as DNA-PAINT | yes (minimizable through adequate buffer conditions) |
| buffer requirements | simple buffer (can be tuned to reduce bleaching of bound imager and binding site depletion) | same as DNA-PAINT | buffer with oxygen-scavenging enzymes and reducing agent required. Buffer composition and conditions need to be adjusted to the fluorophore and sample requirements |

**Extended Data Fig. 1 | Comparison of current DNA-PAINT, DNA-PAINT MINFLUX and MINFLUX implementations.** The three techniques are compared with respect to their state-switching mechanisms, their localization concepts and key performance parameters.

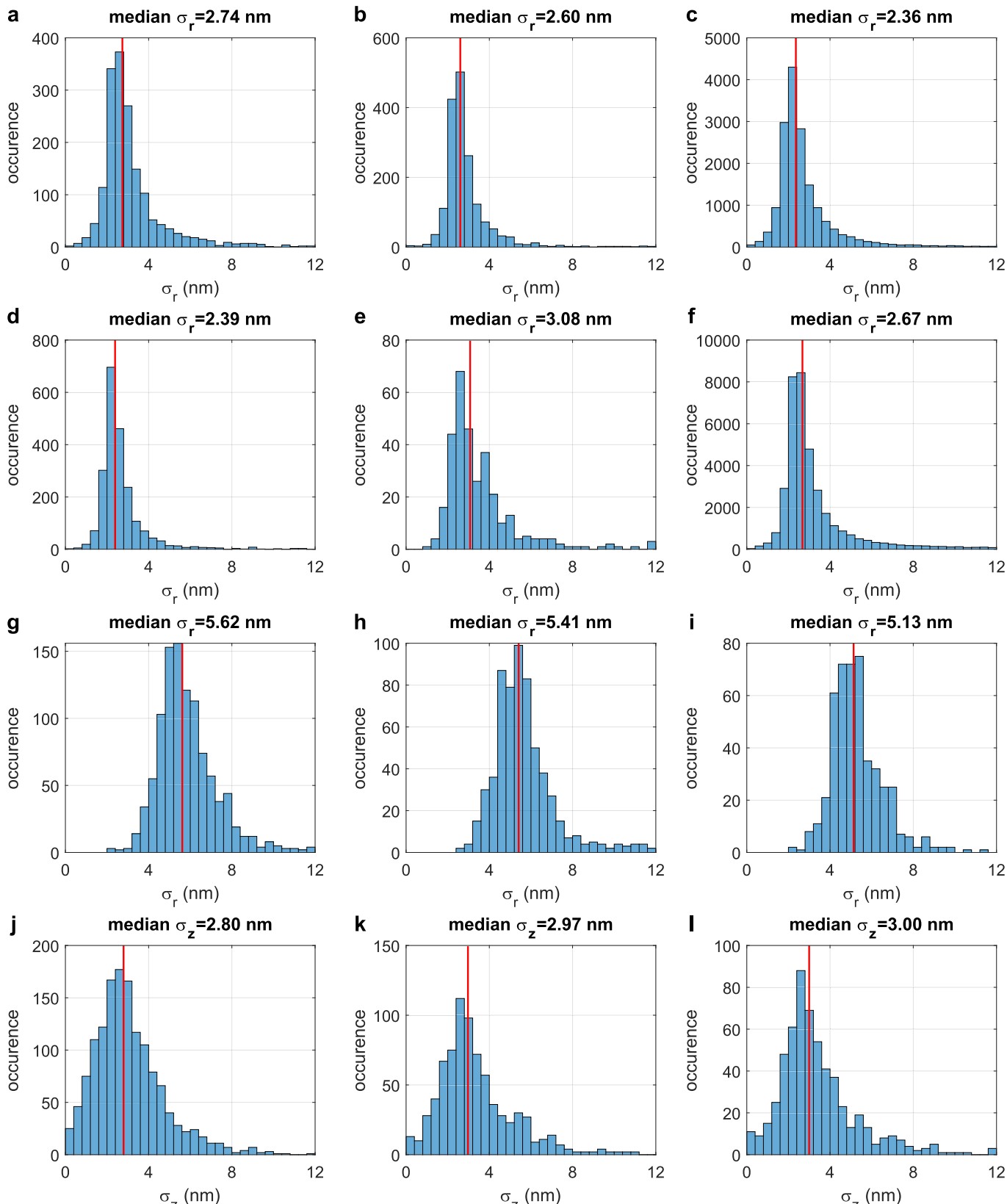

**Extended Data Fig. 2 | Histograms of the localization precisions in Fig. 1 and Fig. 2.** Blue columns represent the frequencies of localization precisions in the given dataset (**a**: Fig. 1a; **b**: Fig. 1b; **c**: Fig. 1c; **d**: Fig. 1d; **e**: Fig. 1e; **f**: Fig. 1f; **g**: Fig. 2 TOM70 $\sigma_r$; **h**: Fig. 2 Mic60 $\sigma_r$; **i**: Fig. 2 ATP5B $\sigma_r$; **j**: Fig. 2 TOM70 $\sigma_z$; **k**: Fig. 2 Mic60 $\sigma_z$; **l**: Fig. 2 ATP5B $\sigma_z$). The red line represents the median of the localization precisions in the dataset.

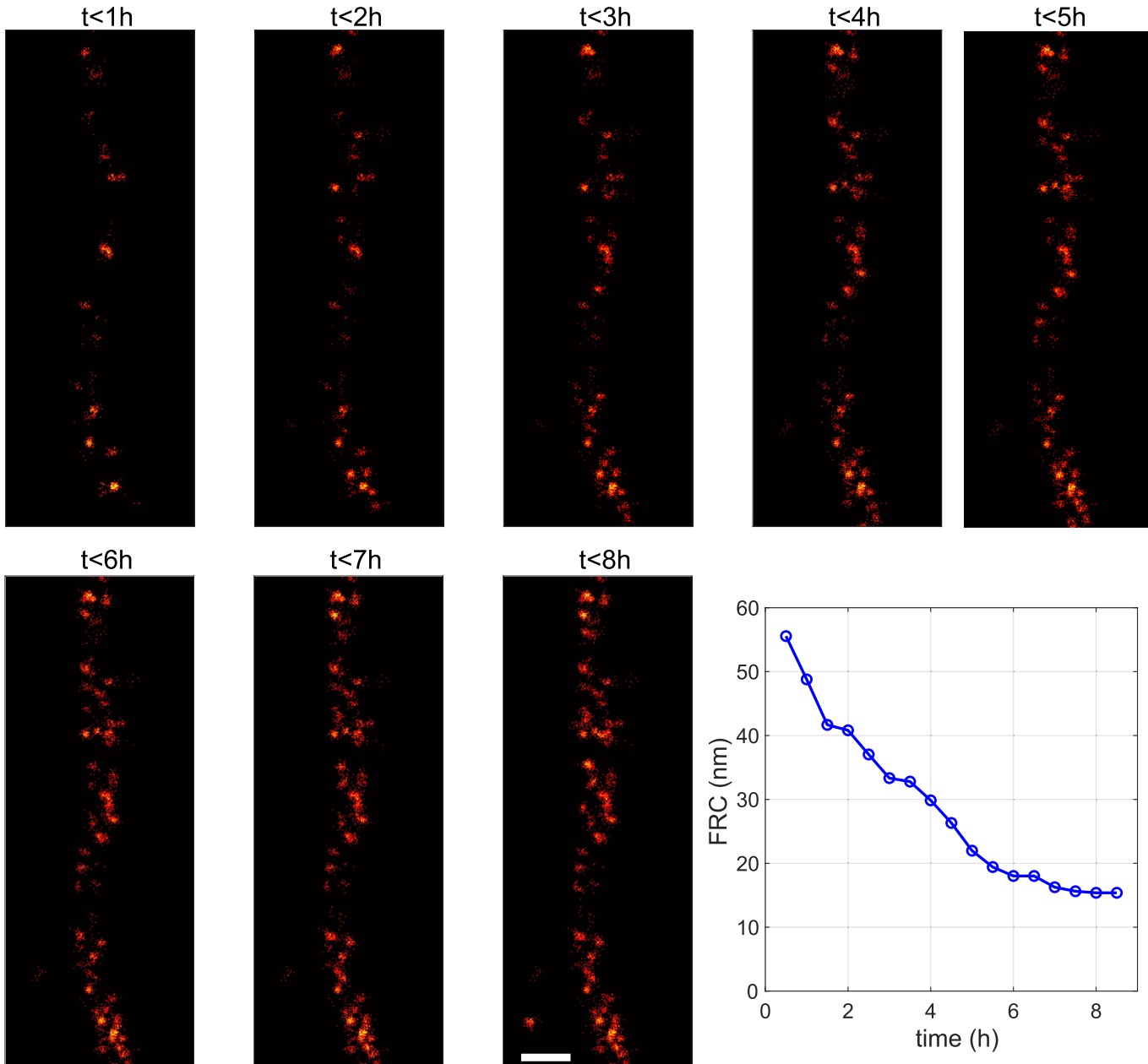

**Extended Data Fig. 3 | The labeling coverage, but not insufficient sampling during a MINFLUX recording, limits the density of localized molecules.**
The individual panels show all recorded localizations in the indicated time intervals. The full field of view of the 8-hour data set is shown in Fig. 1f. The FRC resolution was calculated using all data up to a certain time point (blue circles). After 6-7 h almost no new localizations contribute to the recorded Vimentin filament and the FRC resolution reaches a plateau, suggesting that the imaging time was sufficient to localize the vast majority of available binding sites. Scale bar: 50 nm.

# Reporting Summary

## Statistics

For all statistical analyses, confirm that the following items are present in the figure legend, table legend, main text, or Methods section.

| n/a | Confirmed | |
|---|---|---|
| ☐ | ☒ | The exact sample size ($n$) for each experimental group/condition, given as a discrete number and unit of measurement |
| ☐ | ☒ | A statement on whether measurements were taken from distinct samples or whether the same sample was measured repeatedly |
| ☒ | ☐ | The statistical test(s) used AND whether they are one- or two-sided<br>*Only common tests should be described solely by name; describe more complex techniques in the Methods section.* |
| ☒ | ☐ | A description of all covariates tested |
| ☒ | ☐ | A description of any assumptions or corrections, such as tests of normality and adjustment for multiple comparisons |
| ☐ | ☒ | A full description of the statistical parameters including central tendency (e.g. means) or other basic estimates (e.g. regression coefficient) AND variation (e.g. standard deviation) or associated estimates of uncertainty (e.g. confidence intervals) |
| ☒ | ☐ | For null hypothesis testing, the test statistic (e.g. $F$, $t$, $r$) with confidence intervals, effect sizes, degrees of freedom and $P$ value noted<br>*Give P values as exact values whenever suitable.* |
| ☒ | ☐ | For Bayesian analysis, information on the choice of priors and Markov chain Monte Carlo settings |
| ☒ | ☐ | For hierarchical and complex designs, identification of the appropriate level for tests and full reporting of outcomes |
| ☒ | ☐ | Estimates of effect sizes (e.g. Cohen's $d$, Pearson's $r$), indicating how they were calculated |

*Our web collection on statistics for biologists contains articles on many of the points above.*

## Software and code

Policy information about availability of computer code

| Data collection | Imspector (version 16.3.11647M-devel-win64-MINFLUX, Abberior Instruments) |
|---|---|
| Data analysis | Imspector (version 16.3.11647M-devel-win64-MINFLUX, Abberior Instruments), MATLAB R2018b, https://doi.org/10.5281/zenodo.6563100 |

For manuscripts utilizing custom algorithms or software that are central to the research but not yet described in published literature, software must be made available to editors and reviewers. We strongly encourage code deposition in a community repository (e.g. GitHub). See the Nature Portfolio guidelines for submitting code & software for further information.

## Data

Policy information about availability of data

All manuscripts must include a data availability statement. This statement should provide the following information, where applicable:
- Accession codes, unique identifiers, or web links for publicly available datasets
- A description of any restrictions on data availability
- For clinical datasets or third party data, please ensure that the statement adheres to our policy

All DNA-PAINT MINFLUX localization data have been deposited at https://doi.org/10.5281/zenodo.6563100. The raw data as provided by the microscope software are available at https://doi.org/10.5281/zenodo.6562764

# Field-specific reporting

Please select the one below that is the best fit for your research. If you are not sure, read the appropriate sections before making your selection.

☒ Life sciences ☐ Behavioural & social sciences ☐ Ecological, evolutionary & environmental sciences

For a reference copy of the document with all sections, see nature.com/documents/nr-reporting-summary-flat.pdf

# Life sciences study design

All studies must disclose on these points even when the disclosure is negative.

| | |
|---|---|
| Sample size | No sample-size calculation was performed. The manuscript reports the demonstration of an imaging method, but draws no biological conclusions, and does not examine or compare different biological conditions. This is not a life science study with coparative analyes of a certain sample size |
| Data exclusions | No data was excluded from the analysis |
| Replication | All attempts of replication were successful. All experiments were repeated three or more times with similar results. |
| Randomization | No randomization was performed. This is not a life science study with comparative analyses of biological situations. |
| Blinding | No blinding was performed. There is no comparison of different biological situations performed in this work. |

# Reporting for specific materials, systems and methods

We require information from authors about some types of materials, experimental systems and methods used in many studies. Here, indicate whether each material, system or method listed is relevant to your study. If you are not sure if a list item applies to your research, read the appropriate section before selecting a response.

### Materials & experimental systems

| n/a | Involved in the study |
|---|---|
| ☐ | ☒ Antibodies |
| ☐ | ☒ Eukaryotic cell lines |
| ☒ | ☐ Palaeontology and archaeology |
| ☒ | ☐ Animals and other organisms |
| ☒ | ☐ Human research participants |
| ☒ | ☐ Clinical data |
| ☒ | ☐ Dual use research of concern |

### Methods

| n/a | Involved in the study |
|---|---|
| ☒ | ☐ ChIP-seq |
| ☒ | ☐ Flow cytometry |
| ☒ | ☐ MRI-based neuroimaging |

## Antibodies

| | |
|---|---|
| Antibodies used | anti-Mitofilin (IMMT/Mic60) (10179-1-AP, Proteintech)<br>anti-ATP Synthase Subunit beta ATPB [4.3E8.D10] (ab5432, Abcam)<br>MASSIVE-TAG-Q anti-GFP nanobody (Massive Photonics; no catalogue number available)<br>FluoTag-Q anti-GFP anti-GFP single domain antibody (conjugated with AlexaFluor647) (N0301-AF647-L, NanoTag Biotechnologies)<br><br>IgG anti-rabbit (MASSIVE-AB 1-PLEX, Massive Photonics)<br>IgG anti-mouse (MASSIVE-AB 1-PLEX, Massive Photonics) |
| Validation | anti-Mitofilin (IMMT/Mic60) (10179-1-AP, Proteintech) - we demonstrated the specificity of this antibody with Mitofilin/Mic60 KO cells in a previous publication (Stephan et al, EMBOJ, 2020). anti-ATP Synthase Subunit beta ATPB [4.3E8.D10] (ab5432, Abcam) - this antibody has been tested and used for different applications in various publications (e.g. Steinberg et al, Nat Commun, 2020; Diokmetzidou, J Cell Sci, 2016; Jans et al, PNAS, 2013; etc.), MASSIVE-TAG-Q anti-GFP nanobody (Massive Photonics) and FluoTag-Q anti-GFP single domain antibody (NanoTag Biotechnologies) - both are the same single domain antibodies differently conjugated. This single domain antibody has been tested and used for different applications in various publications (e.g. Sograte-Idrissi et al, Cells, 2019; Oleksiievets et al., Commun Biol, 2022; Seitz et al, Sci Rep, 2019; Thevathasa et al, Nat Methods, 2019) |

## Eukaryotic cell lines

Policy information about cell lines

| | |
|---|---|
| Cell line source(s) | CLS Cell Lines Services GmbH, Eppelheim, Germany (NUP96-mEGFP cell line U2OS-CRISPR-NUP96-mEGFP clone #195 |

| Cell line source(s) | (300174)21 and NUP107-mEGFP cell line HK-2xZFN-mEGFP-Nup107). Cell lines HMGA1-rsEGFP2, Zyxin-rsEGFP2, Vimentin-rsEGFP2 and TOMM70A-Dreiklang were produced from U2OS cells (American Type Culture Collection, Manassas, VA, USA) as described in (Ratz et al, Sci Rep, 2015). |
| --- | --- |
| Authentication | Authentification by microscopy. |
| Mycoplasma contamination | The cell line was tested for mycoplasma contamination and negative results were obtained. |
| Commonly misidentified lines (See ICLAC register) | None. |

