## [Peer Review File · Nature Methods]

Peer Review Information

Manuscript Title: DNA-PAINT MINFLUX Nanoscopy

Corresponding author name(s): Stefan Jokobs

Reviewer Comments & Decisions:

Decision Letter, initial version:

7th Jan 2022

Dear Stefan,

Hello and happy new year!

Your Brief Communication, "DNA-PAINT MINFLUX Nanoscopy", has now been seen by three reviewers. As you will see from their comments below, although the reviewers find your work of considerable potential interest, they have raised a number of concerns. We are interested in the possibility of publishing your paper in Nature Methods, but would like to consider your response to these concerns before we reach a final decision on publication.

We therefore invite you to revise your manuscript to address these concerns. Specifically, we ask that you provide more benchmarking relative to DNA-PAINT and MINFLUX and make necessary code available with the paper alongside addressing the other technical concerns.

* include a point-by-point response to the reviewers and to any editorial suggestions

* please underline/highlight any additions to the text or areas with other significant changes to facilitate review of the revised manuscript

- * address the points listed described below to conform to our open science requirements
- * ensure it complies with our general format requirements as set out in our guide to authors at www.nature.com/naturemethods
- * resubmit all the necessary files electronically by using the link below to access your home page

We hope to receive your revised paper within three months. If you cannot send it within this time, please let us know. In this event, we will still be happy to reconsider your paper at a later date so long as nothing similar has been accepted for publication at Nature Methods or published elsewhere.

OPEN SCIENCE REQUIREMENTS

REPORTING SUMMARY AND EDITORIAL POLICY CHECKLISTS

DATA AVAILABILITY

All novel DNA and RNA sequencing data, protein sequences, genetic polymorphisms, linked genotype and phenotype data, gene expression data, macromolecular structures, and proteomics data must be deposited in a publicly accessible database, and accession codes and associated hyperlinks must be provided in the "Data Availability" section.

Refer to our data policies here: <https://www.nature.com/nature-research/editorial-policies/reporting->

standards#availability-of-data

CODE AVAILABILITY

Please include a "Code Availability" subsection in the Online Methods which details how your custom code is made available. Only in rare cases (where code is not central to the main conclusions of the paper) is the statement "available upon request" allowed (and reasons should be specified).

For more information on our code sharing policy and requirements, please see: <https://www.nature.com/nature-research/editorial-policies/reporting-standards#availability-of-computer-code>

MATERIALS AVAILABILITY

ORCID

Nature Methods is committed to improving transparency in authorship. As part of our efforts in this direction, we are now requesting that all authors identified as 'corresponding author' on published papers create and link their Open Researcher and Contributor Identifier (ORCID) with their account on the Manuscript Tracking System (MTS), prior to acceptance. This applies to primary research papers only. ORCID helps the scientific community achieve unambiguous attribution of all scholarly contributions. You can create and link your ORCID from the home page of the MTS by clicking on 'Modify my Springer Nature account'. For more information please visit please visit www.springernature.com/orcid.

Sincerely,
Rita

Rita Strack, Ph.D.
Senior Editor
Nature Methods

Reviewers' Comments:

Reviewer #1:

Remarks to the Author:

The authors demonstrate MINFLUX with DNA-PAINT. The main difference concerning the earlier work is how the blinking needed for MINFLUX is achieved. DNA-PAINT facilitates blinking through the binding and unbinding of DNA oligo. The technical novelty over the state-of-the-art (SOTA) in terms of reconstruction, hardware, and sample preparation does not become clear from the manuscript. However, I can imagine that imaging experiments that take 6 to 7 hours require an extremely stable system which might require additional technical innovation. The authors can make their case for novelty clearer.

After reading the manuscript I also still wonder what the synergy is between DNA-PAINT and MINFLUX. After the original MINFLUX publication, it has been shown that the localization precision can also significantly be increased by combining repeated localization from the same binding site. This approach works very well on data obtained from long DNA-PAINT acquisitions (<https://doi.org/10.1101/752287>). The combination of DNA-PAINT and MINFLUX is synergetic if the localization precision is higher or if the acquisitions would be faster than the SOTA. The authors can make a stronger case for either since most labs do not use DNA-PAINT anymore beyond proof-of-principle experiments.

Comments main text:

1. The comparison with confocal in a 2D sample (fig 1) is understandable (since it can be produced from the same measurement), but lacks comparison with a more SOTA method. SOTA DNA-PAINT in 2D would be TIRF, so the big question is: How does DNA-PAINT MINFLUX compare to DNA-PAINT TIRF with similar CRLB thresholds for filtering? From this maybe the authors can make a quantified prediction of what the performance would be for ROSE, ModLoc, SIMFLUX, and SIMPLE?
2. It would be great if the authors could show multi-color DNA-PAINT over the whole FOV and ideally on a sample that is often used for benchmarking. The SOTA is at least three colors where one is tubulin (others can be e.g. vimentin and clathrin).
3. The figures in the main text lack quantitative results. The authors must add histograms of localization by taking cross-sections, evaluate the localization precision by linking the localisations (and calculate the std) and quantify their reconstructions in terms of the FRC.
4. For future users of the technique it is important that the authors assess what the impact is of varying the pinhole size, modulation contrast, and background on the maximum achievable localization precision?
5. In the main text, the authors state that imaging experiments longer than 6-7 hours did not add anything. It is not clear if this is because of the accumulation of the drift error, which I expect to incrementally increase, or because of saturation of the FRC i.e. in terms of localization precision and localization density. It would be great if the authors can quantify this because it will give future users insight into what kind of sample can be used for this approach and how long the experiment will take.
6. The authors state that DNA-PAINT MINFLUX has major advantages over dSTORM MINFLUX. It would be essential that the authors show quantitatively how dSTORM MINFLUX compares to DNA-PAINT MINFLUX over such a large FOV. It would be beneficial for future users to see the advantage is, since DNA-PAINT will require extra effort for many labs.

Comments supplement:

7. On a similar note, at various places in main and supplement the localization precision is mentioned but undefined. Is it calculated from the CRLB? Furthermore, the CRLB can be highly biased due to differences in excitation PSF and other factors, for example, the model not matching experiments anymore due to higher background, as mentioned in supplement line 352. It will be necessary for the authors to present a detailed assessment of these experimental factors and present the estimated CRLB as a distribution over the experiment.
8. In the supplement line 301, 304, 425: The relative laser power of 14% seems strange to include, as it is specific to the device. It would be better to stick to absolute measurements and include an estimate of power density at the confocal spot. This can be measured with a power meter from Thorlabs.

Reviewer #2:

Remarks to the Author:

This well written manuscript by Ostersehlt et al describes a combination of MINFLUX, a next

generation super-resolution fluorescence imaging method, with DNA-PAINT, a concept for single-molecule localization based super-resolution microscopy, building on transient binding of fluorescent molecules to the target molecules to be imaged. The motivation of this combined concept, and the synergies which come with it, are convincingly and clearly described. The combined concept, DNA-PAINT MINFLUX Nanoscopy, is applied on several different cellular samples, where the specific advantages of the concept, such as its abilities for 3D imaging, imaging of densely packed molecules and multiplexing (by subsequently adding, and washing away, different orthogonal strands targeting different target docking strands) are clearly demonstrated. The concept thus represents an important new tool and a significant advance in the field of fluorescence imaging.

To further evaluate the synergies, the authors then investigated how certain key parameters influence the performance, where the performance was assessed based on three variables: i) time between valid events (t_{btw}), ii) center-frequency-ratio (CFR), and iii) localization precision ($\sigma(r)$). This performance evaluation is important and highly relevant for all scientists who want to apply this concept in the future. However, the evaluation would be more useful if the outcome could be presented in somewhat more general and transparent measures. In the evaluation, presented mainly in the SI and supplementary notes, several trends in the graphs essentially reflect specific (but not mentioned) settings of the MINFLUX instrument software used (e.g. Figs SN1a, 1b, 2a, 2b and 3a). Also, for several of the parameters investigated, their optimal settings seem difficult to more generally translate into other experimental conditions. The laser powers should preferably be directly stated in their units in the graphs, not percentages, and it would also be useful to know what excitation intensities they correspond to in the sample. A good imager concentration is concluded to be around 2nM. How much will this concentration depend on the dissociation constant (KD) of the imager strand to the docking strand of the target, and what are the dissociation constants for the different strands used? How would different KDs affect the optimal setting of the other parameters studied, and to what extent will it also be a parameter to consider in the choice of imager concentration, in addition to target binding site density?

In conclusion, this manuscript presents an elegant and useful concept, a significant advance in fluorescence-based cellular imaging. With some clarifications and added information on how the key parameters influence its performance, this manuscript will likely be of large interest and value to scientists in the field of cellular imaging.

Minor points:

- P.5, lines 127-128: change "a single binding event" to "single binding events"?
- SI, p.10, line 298: four fold?

Reviewer #3:

Remarks to the Author:

In this brief communication, two existing approaches--DNA-PAINT & MINFLUX--are integrated to improve the latter. Conventional MINFLUX is limited to imaging two fluorescence channels, but by adopting a DNA hybridization scheme with sequential imaging cycles, this limitation is overcome. The authors demonstrate 3D imaging of three proteins in fixed human cells, although theoretically the number of species that can be imaged is unlimited. The manuscript is well written and fits the scope and readership of Nature Methods, but a more convincing visual and quantitative comparison among MINFLUX, DNA-PAINT, and DNA-PAINT MINFLUX should be included.

Major comments

1. Figure 1 compares (diffraction limited) confocal imaging with DNA-PAINT MINFLUX and the latter

performs better. However, as both DNA-PAINT & MINFLUX individually also outperform confocal imaging, this result was to be expected. To understand what the impact of combining DNA-PAINT with MINFLUX is, a visual comparison between all three--DNA-PAINT only, MINFLUX only, and DNA-PAINT MINFLUX--should be provided. For example, does the integration of MINFLUX with DNA-PAINT lowers the resolution due to the linkage error induced by the DNA docking strand? Do they collect fewer localizations, because the total acquisition time is longer?

2. Performance metrics, such as the resolution, are only reported for DNA-PAINT MINFLUX. For potential future users of DNA-PAINT MINFLUX to make an informed decision on what method would be best for them and showcase how DNA-PAINT MINFLUX exploits a synergistic effect, the authors should include a table/figure with quantitative comparison of DNA-PAINT, MINFLUX, and DNA-PAINT MINFLUX. Metrics such as, number of species/colours that can be imaged, resolution/localization precision, acquisition time, etc. can be included.

3. Throughout the study, a very low imager strand concentration of 0.5 - 2.5 nM is used, whereas most DNA-PAINT studies use around 10 nM. Even with 10 nM, the required acquisition time can already be on the order of hours, and this lengthy acquisition time is a major limitation of DNA-PAINT. The authors here require imaging times of up to 7 hours (P4L113).

3.1. Could the authors elaborate on what implications this has for the potential of DNA-PAINT MINFLUX and what applications are currently within reach (and which are not)?

3.2. Several strategies to reduce the acquisition time have been developed in recent years, such as optimising sequence design, buffer composition, imager strand concentration or used protein-assisted strand preforming. Would the authors briefly discuss which of these strategies might be included in later iterations of DNA-PAINT MINFLUX?

4. After the introduction, the first thing mentioned is: "we first explored the influence of a number of key parameters, such as laser power, confocal pinhole size and imager concentration on MINFLUX imaging with DNA-PAINT. Specifically, we determined the influence of these variables on i) the time t_{btw} between valid events, ii) the center-frequency-ratio (CFR), a filter parameter for localizations during image acquisition⁴, and iii) the localization precision σ ." However, later the analysis of these parameters is reported in the Supplementary and in the main only the final recommended values are provided. If the authors want to place such an emphasis on these parameters, this referee suggests to include a more detailed analysis & substantiation in the main text and mention this parameter analysis in the abstract. Furthermore, in line with an earlier comment, this referee suggests to put the found values for laser power, pinhole size, and imager concentration into context by providing comparative values for DNA-PAINT and/or conventional MINFLUX. If this makes the length of this article not fit Brief Communications, the authors may either not emphasise these parameters or consider submitting revision in the form of Research Article.

Minor comments

1. As Nature Methods wishes its publications to contain a technical description that is adequate for reproduction. Would the authors make code & data directly accessible online (e.g. github) instead of upon request?

2. P2L43: typo in "a transient binding", "a" should be removed.

Author Rebuttal to Initial comments: A

Point-by-point response

Reviewers' Comments:

Reviewer #1:

Remarks to the Author:

The authors demonstrate MINFLUX with DNA-PAINT. The main difference concerning the earlier work is how the blinking needed for MINFLUX is achieved. DNA-PAINT facilitates blinking through the binding and unbinding of DNA oligo. The technical novelty over the state-of-the-art (SOTA) in terms of reconstruction, hardware, and sample preparation does not become clear from the manuscript. However, I can imagine that imaging experiments that take 6 to 7 hours require an extremely stable system which might require additional technical innovation. The authors can make their case for novelty clearer.

After reading the manuscript I also still wonder what the synergy is between DNA-PAINT and MINFLUX. After the original MINFLUX publication, it has been shown that the localization precision can also significantly be increased by combining repeated localization from the same binding site. This approach works very well on data obtained from long DNA-PAINT acquisitions (<https://doi.org/10.1101/752287>). The combination of DNA-PAINT and MINFLUX is synergetic if the localization precision is higher or if the acquisitions would be faster than the SOTA. The authors can make a stronger case for either since most labs do not use DNA-PAINT anymore beyond proof-of-principle experiments.

We thank the referee for the helpful comments to improve the manuscript.

The referee is absolutely right, combining the localizations from the same binding site increases the nominal localization precision. We would like to note that in Fazel et al (<https://doi.org/10.1101/752287>) localizations from different events (in case of DNA-PAINT the repeated binding of the imager strand to the docking strand) are combined to increase the localization precision. In this manuscript we combined individual localizations from a single imager strand while it was bound to the docking strand.

The latter approach was described in Pape et al., 2020 (cit. 7). Typically we combined on average 20 localizations from one binding event of the imager strand.

This fact is stated on page 20 of the supplement: *“Thereby we localized each molecule more than 20 times on average, while the imager strand was bound to the docking strand.”*

Indeed, it would be possible to combine the combined localizations. We prefer to abstain from this, because we believe that the obtained nominal sub-nanometer localization precisions would not be helpful.

In the revised version of the manuscript, we clearly state that it is possible to combine individual localizations of a single binding event.

It reads (line 106, page 4): *“As previously demonstrated, the individual localizations of single binding events can also be combined⁷, resulting in higher nominal localization precisions of 0.6 to 0.9 nm (σ_{rc}) (Supplementary Table 1).”*

We show in Supplementary Table 1 a comparison between the measured localization precisions and the combined localization precisions for all images shown. The combined localization precisions are higher than those reported for classical DNA-PAINT recordings.

In the revised version of the manuscript we took great care to elaborate on the synergies between DNA-PAINT and MINFLUX. To this end, we added an entire new paragraph to the introduction of the manuscript (see line 56, page 2 of the main manuscript). We also added the new Supplementary Fig. 1 that provides a comparison between DNA-PAINT nanoscopy, conventional MINFLUX nanoscopy and DNA-PAINT MINFLUX nanoscopy, and highlights the synergies.

However, we slightly disagree with this reviewer that DNA-PAINT is no longer state-of-the-art. Studies using DNA-PAINT are still reported in reputed journals. For example:

Archan et al., Clathrin packets move in slow axonal transport and deliver functional payloads to synapses. *Neuron*, (2021). <https://doi.org/10.1016/j.neuron.2021.08.016>.

Stehr et al., Tracking single particles for hours via continuous DNA-mediated fluorophore exchange. *Nat Commun* (2021). <https://doi.org/10.1038/s41467-021-24223-4>

Sun et al., The prevalence and specificity of local protein synthesis during neuronal synaptic plasticity. *Sci Adv* (2021). <https://www.science.org/doi/abs/10.1126/sciadv.abj0790>

Geertsema et al., Left-handed DNA-PAINT for improved super-resolution imaging in the nucleus. *Nat Biotechnol* (2021). <https://doi.org/10.1038/s41587-020-00753-y>

Clowsley et al., Repeat DNA-PAINT suppresses background and non-specific signals in optical nanoscopy. *Nat Commun* (2021). <https://doi.org/10.1038/s41467-020-20686-z>

Comments main text:

1. The comparison with confocal in a 2D sample (fig 1) is understandable (since it can be produced from the same measurement), but lacks comparison with a more SOTA method. SOTA DNA-PAINT in 2D would be TIRF, so the big question is: How does DNA-PAINT MINFLUX compare to DNA-PAINT TIRF with similar CRLB thresholds for filtering? From this maybe the authors can make a quantified prediction of what the performance would be for ROSE, ModLoc, SIMFLUX, and SIMPLE?

In this manuscript, we did not apply any post-filtering of the data, as we display all valid obtained localization events. (Please note that all data are deposited at <https://doi.org/10.5281/zenodo.6396988>.)

An expression for the CRLB of MINFLUX nanoscopy has been presented in Balzarotti et al., 2017. As the CRLB value depends on the recording scheme, but not on the

labeling strategy, we expect the same values for standard MINFLUX nanoscopy and DNA-PAINT MINFLUX nanoscopy.

Indeed, DNA-PAINT could also be combined with methods such as ROSE, ModLoc, SIMFLUX, SIMPLE, etc. We fully agree that it would be informative to systematically compare MINFLUX nanoscopy with these and other methods. However, as this Brief Communication is not the first report on MINFLUX, we believe that it is not the adequate platform for such a comparison. In fact, we believe that it would be out of the scope of this manuscript and should perhaps be part of a future review-type manuscript.

2. It would be great if the authors could show multi-color DNA-PAINT over the whole FOV and ideally on a sample that is often used for benchmarking. The SOTA is at least three colors where one is tubulin (others can be e.g. vimentin and clathrin).

We fully agree with the reviewer that imaging at least three colors should be regarded as the state-of-the art. In this manuscript we show, for the first time, three color MINFLUX imaging (Fig. 2).

Using the present implementation of MINFLUX nanoscopy it is just not feasible to record an entire large FOV (e.g. 80 x 80 μm) as it would take days to record such an area. Instead, it is more reasonable to record multiple smaller areas, as shown in the manuscript.

Although the combined imaging of tubulin, vimentin and clathrin may be regarded as state-of-the-art for benchmarking many imaging modalities, we believe that these cellular targets are not optimally suited to evaluate the power of MINFLUX nanoscopy: In a cell these structures are generally so far apart from each other that we just do not need MINFLUX nanoscopy for separating them. Therefore, we suggest that three different proteins within the narrow confined spaces of an organelle are much more challenging to record; consequently, we imaged three different proteins in a single mitochondrion (Fig. 2). We believe that this should be regarded as the state-of-the-art for this kind of nanoscopy.

3. The figures in the main text lack quantitative results. The authors must add histograms of localization by taking cross-sections, evaluate the localization precision by linking the localisations (and calculate the std) and quantify their reconstructions in terms of the FRC.

We thank the reviewer for raising this point. For the revised version of the manuscript the localization precisions for all images shown in the manuscript are reported in the Supplementary Table 1. As suggested by the reviewer, in the revised version of the manuscript we show histograms of the distribution of localization precisions (new Suppl. Fig. 2). Please note that the localization precisions were determined by calculating the standard deviation of all localizations with the same TID. The experimental details for this calculation are provided in the Supplementary Methods Section “*MINFLUX 2D data analysis/ Quantification*”.

We believe that the determination of the localization precision of every individual localization event is the most direct and objective approach to provide quantitative information on the localization precision in the images. We consider Fourier ring correlation (FRC) as a less straightforward measure to determine the microscope’s optical resolution abilities, because it is strongly influenced by the label density, which

varies from sample to sample. Also, binding sites that are recorded only once do not meaningfully contribute to the FRC, which requires two independent data sets. Hence we are convinced that providing a general FRC analysis of the data in the manuscript would provide little benefit to the reader and therefore we prefer not to show this analysis in the manuscript.

However, we picked up the suggestion of this reviewer to evaluate the possibility to use the FRC value as a criterion to abort a MINFLUX measurement. Concretely, we determined the FRC for the vimentin recording shown in Supplementary Fig. 2 at different time points of the measurement. We found that the visual impression, namely that after 6-7 hours of MINFLUX imaging no further improvement is visible, is fully confirmed by the FRC determination. After 6-7 hours the FRC value reaches a plateau. This can be used as an abort criterion. Consequently, we added this finding to the manuscript (see new FRC-panel in Supplementary Fig. 3) and discuss the use of FRC as a practical criterion to stop a MINFLUX recording.

Nonetheless, we calculated the FRC values for all images shown in the manuscript (see Table for Referee 1, below).

The FRC on single (non-combined) localization sets are strictly proportional to the estimated localization precisions. This is expected, due to the large number of single localizations per event (typically > 10) dominating the Fourier correlations. A more meaningful analysis is the determination of the FRC of the combined localizations. These values are given in the table below.

	FRC resolution
Figure 1a	8.9 nm
Figure 1b	4.5 nm
Figure 1c	7.6 nm
Figure 1d	10.4 nm
Figure 1e	5.6 nm
Figure 1f	15.2 nm
Figure 2 TOM70	17.2 nm
Figure 2 Mic60	25.6 nm
Figure 2 ATP5B	35.7 nm

4. For future users of the technique it is important that the authors assess what the impact is of varying the pinhole size, modulation contrast, and background on the maximum achievable localization precision?

In the previous version of the manuscript we systematically investigated the influence of the pinhole size, the laser power (which is related to the modulation contrast of the excitation doughnut) and the imager concentration on various parameters, including the localization precision.

We thank the reviewer for suggesting to add the background fluorescence as a parameter. For the revision we added four new panels to the Supplementary Notes (Suppl. Note Fig. I d, Suppl. Note Fig. II d, Suppl. Note Fig. IV d), that report on the

influence of the pinhole size, the laser power and the imager concentration on the background (f_{bg}). We agree that this is a very useful additional data set.

In addition, we added an additional paragraph to the Supplementary Information which puts this systematic analysis into context (pages 19-20).

5. In the main text, the authors state that imaging experiments longer than 6-7 hours did not add anything. It is not clear if this is because of the accumulation of the drift error, which I expect to incrementally increase, or because of saturation of the FRC i.e. in terms of localization precision and localization density. It would be great if the authors can quantify this because it will give future users insight into what kind of sample can be used for this approach and how long the experiment will take.

Indeed, we observed in the experiments shown in Fig. 1f and Supplementary Fig. 3 that after 6-7 hours no additional localization events were recorded. This is not due to drift, as the microscope is very well drift corrected, and we additionally corrected for the remaining drift (explained in the methods section).

The referee is correct in assuming that the FRC saturates after 6-7 hours. We quantified this and added the FRC data to a new panel in Supplementary Fig. 3. We conclude that the FRC may be used as an abort criterion to stop long-term MINFLUX recordings.

This conclusion has also been added to the main text.

It reads (line 122, page 4) „ *This impression was fully in line with a Fourier ring correlation (FRC) analysis¹⁹ of the images recorded at the different time points. After 6-7 hours, the FRC resolution value reached a plateau (Supplementary Fig. 3). We conclude that most of the accessible binding sites had been captured, and that a prolongation of the recording time would not have improved the recording further. We also note that the progression of the FRC resolution values could be used as an abort criterion to determine the endpoint of DNA-PAINT MINFLUX recordings.* ”

6. The authors state that DNA-PAINT MINFLUX has major advantages over dSTORM MINFLUX. It would be essential that the authors show quantitatively how dSTORM MINFLUX compares to DNA-PAINT MINFLUX over such a large FOV. It would be beneficial for future users to see the advantage is, since DNA-PAINT will require extra effort for many labs.

Here, we kindly disagree with the reviewer. To our experience, DNA-PAINT MINFLUX requires no extra efforts compared to dSTORM MINFLUX. In fact, from a practical perspective, DNA-PAINT MINFLUX is easier to use: No complex buffers are required, no bleaching, all components are commercially available, multiplexing is easily achieved, and it is easily adaptable to different target densities.

To explain these advantages better, we added the new Supplementary Fig. 1 to the manuscript. The figure summarizes the differences between DNA-PAINT nanoscopy, DNA-PAINT MINFLUX nanoscopy, and dSTORM MINFLUX nanoscopy.

In addition, we re-wrote parts of the main manuscript and added a paragraph to explain these advantages and the synergies (line 56, page 2).

Comments supplement:

7. On a similar note, at various places in main and supplement the localization precision is mentioned but undefined. Is it calculated from the CRLB? Furthermore, the CRLB can be highly biased due to differences in excitation PSF and other factors, for example, the model not matching experiments anymore due to higher background, as mentioned in supplement line 352. It will be necessary for the authors to present a detailed assessment of these experimental factors and present the estimated CRLB as a distribution over the experiment.

Throughout the manuscript, the localization precision has not been calculated, but experimentally determined from consecutive localizations during a single binding event. This is indeed an advantage, as the CRLB is not required for the determination of the localizations precision.

Supplementary Table 1 and the new Supplementary Fig. 2 report on the experimentally determined spread of the localization precisions. These values do not require any assumption on the excitation PSF or the background level.

A detailed explanation for the determination of the localization precision is provided in the revised Methods sections. It reads (Supplementary Methods / MINFLUX data analysis, page 5):

“To estimate the localization precision of a measurement as the third quantification parameter, the standard deviation σ_{rr} was calculated for each molecule (at least 5 localizations with the same exported parameter TID) as $\sigma_{rr} = \sqrt{\sigma_x^2 + \sigma_y^2}$ with the standard deviations of the xx- and yy- coordinates as determined by the microscope (exported parameter POS). The median σ_{rr} represents the stated localization precision. The combined localization precision was estimated as $\sigma_{rrrr} = \langle \langle \sigma_{rr} \rangle / \sqrt{nn} \rangle_{mm}$, i.e. the weighted average of the average single localization precision σ_{rr} divided by \sqrt{nn} and weighted by the occurrence of nn being the number of single localizations with the same TID.”

8. In the supplement line 301, 304, 425: The relative laser power of 14% seems strange to include, as it is specific to the device. It would be better to stick to absolute measurements and include an estimate of power density at the confocal spot. This can be measured with a power meter from Thorlabs.

We thank the reviewer for this suggestion. In the revised version of the manuscript, all laser powers are reported as μW deposited in the sample.

Reviewer #2:

Remarks to the Author:

This well written manuscript by Ostersehl et al describes a combination of MINFLUX, a next generation super-resolution fluorescence imaging method, with DNA-PAINT, a concept for single-molecule localization based super-resolution microscopy, building on transient binding of fluorescent molecules to the target molecules to be imaged. The motivation of this combined concept, and the synergies which come with it, are convincingly and clearly described. The combined concept, DNA-PAINT MINFLUX Nanoscopy, is applied on several different cellular samples, where the specific advantages of the concept, such as its abilities for 3D imaging, imaging of densely packed molecules and multiplexing (by subsequently adding, and washing away, different orthogonal strands targeting different target docking strands) are clearly demonstrated. The concept thus represents an important new tool and a significant advance in the field of fluorescence imaging.

We thank the reviewer for the positive view on our manuscript.

To further evaluate the synergies, the authors then investigated how certain key parameters influence the performance, where the performance was assessed based on three variables: i) time between valid events ($t(\text{btw})$), ii) center-frequency-ratio (CFR), and iii) localization precision ($\sigma(r)$). This performance evaluation is important and highly relevant for all scientists who want to apply this concept in the future. However, the evaluation would be more useful if the outcome could be presented in somewhat more general and transparent measures. In the evaluation, presented mainly in the SI and supplementary notes, several trends in the graphs essentially reflect specific (but not mentioned) settings of the MINFLUX instrument software used (e.g. Figs SN1a, 1b, 2a, 2b and 3a). Also, for several of the parameters investigated, their optimal settings seem difficult to more generally translate into other experimental conditions.

We fully agree that it is a difficult balance between a more general description of the evaluation of the MINFLUX parameters and a description of the specific settings tailored to the microscope used.

Because the microscope used is the only MINFLUX system available on the market and because it is a new and largely untested technology, we believe that it is beneficial for the readers to have information also on specific settings. In the revised manuscript, all settings are detailed in the full MINFLUX imaging sequence given in Supplementary data set 1. Key parameters of the MINFLUX sequence are now pointed out in the paragraph *Supplementary Methods/MINFLUX sequences* (page 7). Practically, information on these settings may help to set up experiments and therefore we prefer to keep information on these specific settings in the Supplemental Information.

In order to provide an additional more general parameter that can be used to determine the performance of a MINFLUX microscope, we report in the revised manuscript additionally on the measured background fluorescence. For the revision, we added three new panels to the Supplementary Notes (Suppl. Note Fig. I d, Suppl. Note Fig. II d, Suppl. Note Fig. IV d), that report on the influence of the pinhole size, the laser power and the imager concentration on the background (f_{bg}).

In addition, we added a paragraph to the Supplementary Information which puts this systematic analysis into a more general context (pages 19-20).

In order to allow readers to analyze the data themselves and to be as transparent as possible, we not only included the entire MINFLUX sequence to the Supplementary Information, but also uploaded the entire analysis software-suite including all localization data (<https://doi.org/10.5281/zenodo.6396988>).

The laser powers should preferably be directly stated in their units in the graphs, not percentages, and it would also be useful to know what excitation intensities they correspond to in the sample.

We thank the reviewer for this suggestion. In the revised version of the manuscript, all laser powers are reported as μW deposited in the sample.

A good imager concentration is concluded to be around 2nM. How much will this concentration depend on the dissociation constant (KD) of the imager strand to the docking strand of the target, and what are the dissociation constants for the different strands used? How would different KDs affect the optimal setting of the other parameters studied, and to what extent will it also be a parameter to consider in the choice of imager concentration, in addition to target binding site density?

Yes, absolutely, the concentration will depend on the KD of the imager strand to the docking strand. Unfortunately, we do not know the KD, as the manufacturer of these strands (Massive Photonics, Graefeling, Germany) does not provide information on their sequence or their KD.

The effects of different KDs on the imaging parameters in DNA-PAINT nanoscopy have been investigated previously. Many of these findings can be translated to DNA-PAINT MINFLUX nanoscopy. To address the issues raised by the reviewer, we added an entire new paragraph (*Possible further improvements*) to the end of the Supplementary Notes (see page 20), providing additional references and discussing strategies to modify the imaging parameters by using other imager and docking strands as well as further modifications in using PAINT.

In conclusion, this manuscript presents an elegant and useful concept, a significant advance in fluorescence-based cellular imaging. With some clarifications and added information on how the key parameters influence its performance, this manuscript will likely be of large interest and value to scientists in the field of cellular imaging.

Thank you for your helpful and supportive comments.

Minor points:

- P.5, lines 127-128: change “a single binding event” to “single binding events”?

Done.

- SI, p.10, line 298: four fold?

Indeed, the text was misleading. The increase is in fact six-fold, because at the last iteration of the sequence, the power is six times higher than in the first iteration.

We clarified this in the text.

This is now stated in the legend to Supplementary Note Figure I.

In addition, we now explain the power increase with each iteration in a new table in the paragraph *Supplementary Methods/MINFLUX sequences* (page 7).

Reviewer #3:

Remarks to the Author:

In this brief communication, two existing approaches--DNA-PAINT & MINFLUX--are integrated to improve the latter. Conventional MINFLUX is limited to imaging two fluorescence channels, but by adopting a DNA hybridization scheme with sequential imaging cycles, this limitation is overcome. The authors demonstrate 3D imaging of three proteins in fixed human cells, although theoretically the number of species that can be imaged is unlimited. The manuscript is well written and fits the scope and readership of Nature Methods, but a more convincing visual and quantitative comparison among MINFLUX, DNA-PAINT, and DNA-PAINT MINFLUX should be included.

We thank this reviewer for the positive view on our manuscript and the helpful suggestions. We provide an extensive comparison of DNA-PAINT nanoscopy, conventional (dSTORM) MINFLUX nanoscopy, and DNA-PAINT MINFLUX nanoscopy, as detailed in the answer below.

Major comments

1. Figure 1 compares (diffraction limited) confocal imaging with DNA-PAINT MINFLUX and the latter performs better. However, as both DNA-PAINT & MINFLUX individually also outperform confocal imaging, this result was to be expected. To understand what the impact of combining DNA-PAINT with MINFLUX is, a visual comparison between all three--DNA-PAINT only, MINFLUX only, and DNA-PAINT MINFLUX--should be provided. For example, does the integration of MINFLUX with DNA-PAINT lower the resolution due to the linkage error induced by the DNA docking strand? Do they collect fewer localizations, because the total acquisition time is longer?

We thank the reviewer for this suggestion. To address this point we have added an entire new Figure (Supplementary Fig. 1). It provides a detailed comparison of DNA-PAINT nanoscopy, conventional (dSTORM) MINFLUX nanoscopy, and DNA-PAINT MINFLUX nanoscopy. The figure details differences between the methods with regard to several performance parameters. An experimental side-by-side comparison of the methods would be out of the scope of this Brief Communication.

In addition, in the revised main manuscript text we now detail the synergistic impact of combining DNA-PAINT with MINFLUX recordings (line 56, page 2).

2. Performance metrics, such as the resolution, are only reported for DNA-PAINT MINFLUX. For potential future users of DNA-PAINT MINFLUX to make an informed decision on what method would be best for them and showcase how DNA-PAINT MINFLUX exploits a synergistic effect, the authors should include a table/figure with quantitative comparison of DNA-PAINT, MINFLUX, and DNA-PAINT MINFLUX. Metrics such as, number of species/colours that can be imaged, resolution/localization precision, acquisition time, etc. can be included.

In the revised manuscript, the new Supplementary Fig. 1 explicitly mentions metrics such as number of colors, attainable localization precision, but also limitations such as the requirement for specific buffer conditions or the need for specific illumination

schemes. We believe that this matrix supports an informed decision on the choice of a suitable method to experimentally address a specific question.

We have also added a short paragraph to the main text describing the synergies achieved by combining DNA-PAINT and MINFLUX.

It reads (Page 2):

“We reasoned that by combining DNA-PAINT with MINFLUX recording, we could synergistically benefit from the advantages of both methods. As in the current MINFLUX nanoscopy implementations, the ‘background’ fluorescence stemming from diffusing imager strands is suppressed by the confocal pinhole, DNA-PAINT MINFLUX nanoscopy can be used in the far-field mode. DNA-PAINT MINFLUX nanoscopy is expected to provide the same single-digit nanometer resolution as conventional MINFLUX nanoscopy. Because the state-switching kinetics are determined by the binding of an imager strand to a docking strand, no dedicated buffer systems are required, and the kinetics can be adapted to the density of the targets by tuning the imager concentration. As in conventional MINFLUX nanoscopy using photoswitchable dyes, also in DNA-PAINT MINFLUX nanoscopy the individual localizations are recorded one-by-one. Thus the imaging time scales with the number of targets, making single-beam scanning MINFLUX particularly suited for recording small regions of interest. Another intrinsic benefit of using PAINT is the fact that when densely packed molecules are imaged, successive fluorophore docking avoids the interaction of fluorophores belonging to neighboring target molecules. Hence co-activation and mutual fluorophore quenching is largely avoided. Finally, as multiple orthogonal imager strands can be applied sequentially, each binding to a different docking strand (Exchange DNA-PAINT)¹⁷, addressing multiple targets should also be straightforward. For an overview of synergies, see also Supplementary Fig. 1.”

Of course, it would be possible to go more into detail, but we believe that a more detailed comparison would be better suited for a future review than for a Brief Communication.

3. Throughout the study, a very low imager strand concentration of 0.5 - 2.5 nM is used, whereas most DNA-PAINT studies use around 10 nM. Even with 10 nM, the required acquisition time can already be on the order of hours, and this lengthy acquisition time is a major limitation of DNA-PAINT. The authors here require imaging times of up to 7 hours (P4L113).

The reviewer is right, the current implementation of MINFLUX is inherently slow for larger fields of view. This is clearly stated on several occasions throughout the manuscript. By modifying the MINFLUX sequence, but also by adapting the labeling strategy, there are options to speed up the imaging process within certain limits. This is discussed in a new paragraph at the end of the Supplementary Notes.

3.1. Could the authors elaborate on what implications this has for the potential of DNA-PAINT MINFLUX and what applications are currently within reach (and which are not)?

Thank you for raising this point. MINFLUX, and DNA-PAINT MINFLUX is particularly suited for small ROIs, rather than for whole cells. This is now stated in the main manuscript.

It reads (line 64, page 3):

“As in conventional MINFLUX nanoscopy using photoswitchable dyes, also in DNA-PAINT MINFLUX nanoscopy the individual localizations are recorded one-by-one. Thus the imaging time scales with the number of targets, making single-beam scanning MINFLUX particularly suited for recording small regions of interest.”

3.2. Several strategies to reduce the acquisition time have been developed in recent years, such as optimising sequence design, buffer composition, imager strand concentration or used protein-assisted strand preforming. Would the authors briefly discuss which of these strategies might be included in later iterations of DNA-PAINT MINFLUX?

Thank you for this suggestion. For the revised version of the manuscript, we added a new paragraph to the Supplemental Notes (*Possible further improvements*) that discusses options to speed up the imaging process. This includes other PAINT variants (Förster resonance energy transfer (FRET)-based probes, caged, photo-activatable dyes, fluorogenic DNA-PAINT probes, preloading of DNA-PAINT imager strands with Argonaute proteins, and improved sequences), modifications in the MINFLUX recording sequence, and ultimately parallelization.

4. After the introduction, the first thing mentioned is: “we first explored the influence of a number of key parameters, such as laser power, confocal pinhole size and imager concentration on MINFLUX imaging with DNA-PAINT. Specifically, we determined the influence of these variables on i) the time btw between valid events, ii) the center-frequency-ratio (CFR), a filter parameter for localizations during image acquisition⁴, and iii) the localization precision σ .” However, later the analysis of these parameters is reported in the Supplementary and in the main only the final recommended values are provided. If the authors want to place such an emphasis on these parameters, this referee suggests to include a more detailed analysis & substantiation in the main text and mention this parameter analysis in the abstract. Furthermore, in line with an earlier comment, this referee suggests to put the found values for laser power, pinhole size, and imager concentration into context by providing comparative values for DNA-PAINT and/or conventional MINFLUX. If this makes the length of this article not fit Brief Communications, the authors may either not emphasise these parameters or consider submitting revision in the form of Research Article.

We do see the point raised by the reviewer. We re-wrote the introduction to reduce the emphasis on the analysis of the parameters. The main manuscript is shorter and more legible due to this modification.

We believe that the reported data fit best to the Brief Communications format, and rather would prefer not to inflate the manuscript to a full Research Article. Therefore, we prefer to follow the suggestion of the reviewer not to emphasize these parameters in order to keep the paper in a short and compact format.

Minor comments

1. As Nature Methods wishes its publications to contain a technical description that is adequate for reproduction. Would the authors make code & data directly accessible online (e.g. github) instead of upon request?

Yes, all code and all data will be made available via zenodo.org. Concretely, a comprehensive software package (written in Matlab) for drift correction, precision estimation as well as CFR and FRC calculations will be made accessible online. The software package also includes localization data for all figures presented in the manuscript.

The localization data and all custom codes used for image analysis are available at <https://doi.org/10.5281/zenodo.6396988>.

2. P2L43: typo in “a transient binding”, “a” should be removed.

Done.

Decision Letter, first revision: A

5th Apr 2022

Dear Professor Jakobs,

Thank you for submitting your revised manuscript entitled "DNA-PAINT MINFLUX Nanoscopy". We generally found the revision quite strong. However, we are concerned that sending the current manuscript out to review could lead to unnecessary delays and possibly an undesirable outcome of the review process.

In particular, we would like to see a direct experimental comparison between either DNA-PAINT (dSTORM) and DNA-PAINT MINFLUX -OR- between MINFLUX and DNA-PAINT MINFLUX. Two reviewers thought such experiments would strengthen the paper. For our biologist readers, we want it to be explicitly clear what benefits come either from doing MINFLUX instead of dSTORM if you're already using DNA-PAINT -OR- what benefits come from doing DNA-PAINT labeling if you're already doing MINFLUX.

We therefore invite you to revise your manuscript to address these concerns before we make a final determination on whether to send your manuscript for external peer-review. Please ensure that the revised version is as concise as possible, and that it conforms to our format requirements (see <http://www.nature.com/nmeth> for our Guide to Authors).

We shall hope to receive your revised version as soon as you are able to complete the suggested revisions. If something similar is published in the interim we will have to consider the impact it has on the novelty of the revised manuscript.

If you anticipate a delay of more than four weeks, please let us know. In this event, we will still be happy to reconsider your paper at a later date so long as nothing similar has been accepted for publication at Nature Methods or published elsewhere. In the event of publication, however, the received date would be that of the revised rather than the original version.

If you are not interested in submitting a revised manuscript in the future please let me know immediately so we can close your file. If you have any questions, please contact me.

Please use the link below when you are prepared to resubmit.

Thank you for your interest in Nature Methods.

Sincerely,
Rita

Rita Strack, Ph.D.
Senior Editor
Nature Methods

** For Nature Research Group general information and news for authors, see
<http://npg.nature.com/authors>.

Author Rebuttal, first revision: B

Point-by-point response

Reviewers' Comments:

Reviewer #1:

Remarks to the Author:

The authors demonstrate MINFLUX with DNA-PAINT. The main difference concerning the earlier work is how the blinking needed for MINFLUX is achieved. DNA-PAINT facilitates blinking through the binding and unbinding of DNA oligo. The technical novelty over the state-of-the-art (SOTA) in terms of reconstruction, hardware, and sample preparation does not become clear from the manuscript. However, I can imagine that imaging experiments that take 6 to 7 hours require an extremely stable system which might require additional technical innovation. The authors can make their case for novelty clearer.

After reading the manuscript I also still wonder what the synergy is between DNA-PAINT and MINFLUX. After the original MINFLUX publication, it has been shown that the localization precision can also significantly be increased by combining repeated localization from the same binding site. This approach works very well on data obtained from long DNA-PAINT acquisitions (<https://doi.org/10.1101/752287>). The combination of DNA-PAINT and MINFLUX is synergetic if the localization precision is higher or if the acquisitions would be faster than the SOTA. The authors can make a stronger case for either since most labs do not use DNA-PAINT anymore beyond proof-of-principle experiments.

We thank the referee for the helpful comments to improve the manuscript.

The referee is absolutely right, combining the localizations from the same binding site increases the nominal localization precision. We would like to note that in Fazel et al (<https://doi.org/10.1101/752287>) localizations from different events (in case of DNA-PAINT the repeated binding of the imager strand to the docking strand) are combined to increase the localization precision. In this manuscript we combined individual localizations from a single imager strand while it was bound to the docking strand.

The latter approach was described in Pape et al., 2020 (cit. 7). Typically we combined on average 20 localizations from one binding event of the imager strand.

This fact is stated on page 20 of the supplement: *“Thereby we localized each molecule more than 20 times on average, while the imager strand was bound to the docking strand.”*

Indeed, it would be possible to combine the combined localizations. We prefer to abstain from this, because we believe that the obtained nominal sub-nanometer localization precisions would not be helpful.

In the revised version of the manuscript, we clearly state that it is possible to combine individual localizations of a single binding event.

It reads (line 106, page 4): *“As previously demonstrated, the individual localizations of single binding events can also be combined⁷, resulting in higher nominal localization precisions of 0.6 to 0.9 nm (σ_{rc}) (Supplementary Table 1).”*

We show in Supplementary Table 1 a comparison between the measured localization precisions and the combined localization precisions for all images shown. The combined localization precisions are higher than those reported for classical DNA-PAINT recordings.

In the revised version of the manuscript we took great care to elaborate on the synergies between DNA-PAINT and MINFLUX. To this end, we added an entire new paragraph to the introduction of the manuscript (see line 56, page 2 of the main manuscript). We also added the new Supplementary Fig. 1 that provides a comparison between DNA-PAINT nanoscopy, conventional MINFLUX nanoscopy and DNA-PAINT MINFLUX nanoscopy, and highlights the synergies.

However, we slightly disagree with this reviewer that DNA-PAINT is no longer state-of-the-art. Studies using DNA-PAINT are still reported in reputed journals.

For example:

Archan et al., Clathrin packets move in slow axonal transport and deliver functional payloads to synapses. *Neuron*, (2021). <https://doi.org/10.1016/j.neuron.2021.08.016>.

Stehr et al., Tracking single particles for hours via continuous DNA-mediated fluorophore exchange. *Nat Commun* (2021). <https://doi.org/10.1038/s41467-021-24223-4>

Sun et al., The prevalence and specificity of local protein synthesis during neuronal synaptic plasticity. *Sci Adv* (2021). <https://www.science.org/doi/abs/10.1126/sciadv.abj0790>

Geertsema et al., Left-handed DNA-PAINT for improved super-resolution imaging in the nucleus. *Nat Biotechnol* (2021). <https://doi.org/10.1038/s41587-020-00753-y>

Clowsley et al., Repeat DNA-PAINT suppresses background and non-specific signals in optical nanoscopy. *Nat Commun* (2021). <https://doi.org/10.1038/s41467-020-20686-z>

Comments main text:

1. The comparison with confocal in a 2D sample (fig 1) is understandable (since it can be produced from the same measurement), but lacks comparison with a more SOTA method. SOTA DNA-PAINT in 2D would be TIRF, so the big question is: How does DNA-PAINT MINFLUX compare to DNA-PAINT TIRF with similar CRLB thresholds for filtering? From this maybe the authors can make a quantified prediction of what the performance would be for ROSE, ModLoc, SIMFLUX, and SIMPLE?

In this manuscript, we did not apply any post-filtering of the data, as we display all valid obtained localization events. (Please note that all data are deposited at <https://doi.org/10.5281/zenodo.6396988>.)

An expression for the CRLB of MINFLUX nanoscopy has been presented in Balzarotti et al., 2017. As the CRLB value depends on the recording scheme, but not on the

labeling strategy, the same values for standard MINFLUX nanoscopy and DNA-PAINT MINFLUX nanoscopy are to be expected. Still, we experimentally compared in the revised version of the manuscript DNA-PAINT MINFLUX and dSTORM MINFLUX (Fig. 1d and Fig. 1e). We achieved the same image quality and similar localization precisions (Supplementary Table 1).

Indeed, DNA-PAINT could also be combined with methods such as ROSE, ModLoc, SIMFLUX, SIMPLE, etc. We fully agree that it would be informative to systematically compare MINFLUX nanoscopy with these and other methods. However, as this Brief Communication is not the first report on MINFLUX, we believe that it is not the adequate platform for such a comparison. In fact, we believe that it would be out of the scope of this manuscript and should perhaps be part of a future review-type manuscript.

2. It would be great if the authors could show multi-color DNA-PAINT over the whole FOV and ideally on a sample that is often used for benchmarking. The SOTA is at least three colors where one is tubulin (others can be e.g. vimentin and clathrin).

We fully agree with the reviewer that imaging at least three colors should be regarded as the state-of-the art. In this manuscript we show, for the first time, three color MINFLUX imaging (Fig. 2).

Using the present implementation of MINFLUX nanoscopy it is just not feasible to record an entire large FOV (e.g. 80 x 80 μm) as it would take days to record such an area. Instead, it is more reasonable to record multiple smaller areas, as shown in the manuscript.

Although the combined imaging of tubulin, vimentin and clathrin may be regarded as state-of-the-art for benchmarking many imaging modalities, we believe that these cellular targets are not optimally suited to evaluate the power of MINFLUX nanoscopy: In a cell these structures are generally so far apart from each other that we just do not need MINFLUX nanoscopy for separating them. Therefore, we suggest that three different proteins within the narrow confined spaces of an organelle are much more challenging to record; consequently, we imaged three different proteins in a single mitochondrion (Fig. 2). We believe that this should be regarded as the state-of-the-art for this kind of nanoscopy.

3. The figures in the main text lack quantitative results. The authors must add histograms of localization by taking cross-sections, evaluate the localization precision by linking the localisations (and calculate the std) and quantify their reconstructions in terms of the FRC.

We thank the reviewer for raising this point. For the revised version of the manuscript the localization precisions for all images shown in the manuscript are reported in the Supplementary Table 1. As suggested by the reviewer, in the revised version of the manuscript we show histograms of the distribution of localization precisions (new Suppl. Fig. 2). Please note that the localization precisions were determined by calculating the standard deviation of all localizations with the same TID. The experimental details for this calculation are provided in the Supplementary Methods Section “*MINFLUX 2D data analysis/ Quantification*”.

We believe that the determination of the localization precision of every individual localization event is the most direct and objective approach to provide quantitative information on the localization precision in the images. We consider Fourier ring correlation (FRC) as a less straightforward measure to determine the microscope's optical resolution abilities, because it is strongly influenced by the label density, which varies from sample to sample. Also, binding sites that are recorded only once do not meaningfully contribute to the FRC, which requires two independent data sets. Hence we are convinced that providing a general FRC analysis of the data in the manuscript would provide little benefit to the reader and therefore we prefer not to show this analysis in the manuscript.

However, we picked up the suggestion of this reviewer to evaluate the possibility to use the FRC value as a criterion to abort a MINFLUX measurement. Concretely, we determined the FRC for the vimentin recording shown in Supplementary Fig. 2 at different time points of the measurement. We found that the visual impression, namely that after 6-7 hours of MINFLUX imaging no further improvement is visible, is fully confirmed by the FRC determination. After 6-7 hours the FRC value reaches a plateau. This can be used as an abort criterion. Consequently, we added this finding to the manuscript (see new FRC-panel in Supplementary Fig. 3) and discuss the use of FRC as a practical criterion to stop a MINFLUX recording.

Nonetheless, we calculated the FRC values for all images shown in the manuscript (see Table for Referee 1, below).

The FRC on single (non-combined) localization sets are strictly proportional to the estimated localization precisions. This is expected, due to the large number of single localizations per event (typically > 10) dominating the Fourier correlations. A more meaningful analysis is the determination of the FRC of the combined localizations. These values are given in the table below.

	FRC resolution
Figure 1a	8.9 nm
Figure 1b	4.5 nm
Figure 1c	7.6 nm
Figure 1d	10.4 nm
Figure 1e	5.6 nm
Figure 1f	15.2 nm
Figure 2 TOM70	17.2 nm
Figure 2 Mic60	25.6 nm
Figure 2 ATP5B	35.7 nm

4. For future users of the technique it is important that the authors assess what the impact is of varying the pinhole size, modulation contrast, and background on the maximum achievable localization precision?

In the previous version of the manuscript we systematically investigated the influence of the pinhole size, the laser power (which is related to the modulation contrast of the

excitation doughnut) and the imager concentration on various parameters, including the localization precision.

We thank the reviewer for suggesting to add the background fluorescence as a parameter. For the revision we added four new panels to the Supplementary Notes (Suppl. Note Fig. I d, Suppl. Note Fig. II d, Suppl. Note Fig. IV d), that report on the influence of the pinhole size, the laser power and the imager concentration on the background (f_{bg}). We agree that this is a very useful additional data set.

In addition, we added an additional paragraph to the Supplementary Information which puts this systematic analysis into context (pages 19-20).

5. In the main text, the authors state that imaging experiments longer than 6-7 hours did not add anything. It is not clear if this is because of the accumulation of the drift error, which I expect to incrementally increase, or because of saturation of the FRC i.e. in terms of localization precision and localization density. It would be great if the authors can quantify this because it will give future users insight into what kind of sample can be used for this approach and how long the experiment will take.

Indeed, we observed in the experiments shown in Fig. 1f and Supplementary Fig. 3 that after 6-7 hours no additional localization events were recorded. This is not due to drift, as the microscope is very well drift corrected, and we additionally corrected for the remaining drift (explained in the methods section).

The referee is correct in assuming that the FRC saturates after 6-7 hours. We quantified this and added the FRC data to a new panel in Supplementary Fig. 3. We conclude that the FRC may be used as an abort criterion to stop long-term MINFLUX recordings.

This conclusion has also been added to the main text.

It reads (line 125, page 5) „ *This impression was fully in line with a Fourier ring correlation (FRC) analysis¹⁹ of the images recorded at the different time points. After 6-7 hours, the FRC resolution value reached a plateau (Supplementary Fig. 3). We conclude that most of the accessible binding sites had been captured, and that a prolongation of the recording time would not have improved the recording further. We also note that the progression of the FRC resolution values could be used as an abort criterion to determine the endpoint of DNA-PAINT MINFLUX recordings.* ”

6. The authors state that DNA-PAINT MINFLUX has major advantages over dSTORM MINFLUX. It would be essential that the authors show quantitatively how dSTORM MINFLUX compares to DNA-PAINT MINFLUX over such a large FOV. It would be beneficial for future users to see the advantage is, since DNA-PAINT will require extra effort for many labs.

Here, we kindly disagree with the reviewer. To our experience, DNA-PAINT MINFLUX requires no extra efforts compared to dSTORM MINFLUX. In fact, from a practical perspective, DNA-PAINT MINFLUX is easier to use: No complex buffers are required, no bleaching, all components are commercially available, multiplexing is easily achieved, and it is easily adaptable to different target densities.

To explain these advantages better, we added the new Supplementary Fig. 1 to the manuscript. The figure summarizes the differences between DNA-PAINT nanoscopy, DNA-PAINT MINFLUX nanoscopy, and dSTORM MINFLUX nanoscopy.

We re-wrote parts of the main manuscript and added a paragraph to explain these advantages and the synergies better (line 56, page 2).

To experimentally compare DNA-PAINT MINFLUX with dSTORM MINFLUX we performed additional experiments. Fig. 1d and 1e show a comparison of dSTORM MINFLUX and DNA-PAINT MINFLUX recordings of the same cellular structure (Nup96-GFP). The image quality was the same and similar localization precisions were achieved (Supplementary Table 1).

Comments supplement:

7. On a similar note, at various places in main and supplement the localization precision is mentioned but undefined. Is it calculated from the CRLB? Furthermore, the CRLB can be highly biased due to differences in excitation PSF and other factors, for example, the model not matching experiments anymore due to higher background, as mentioned in supplement line 352. It will be necessary for the authors to present a detailed assessment of these experimental factors and present the estimated CRLB as a distribution over the experiment.

Throughout the manuscript, the localization precision has not been calculated, but experimentally determined from consecutive localizations during a single binding event. This is indeed an advantage, as the CRLB is not required for the determination of the localizations precision.

Supplementary Table 1 and the new Supplementary Fig. 2 report on the experimentally determined spread of the localization precisions. These values do not require any assumption on the excitation PSF or the background level.

A detailed explanation for the determination of the localization precision is provided in the revised Methods sections. It reads (Supplementary Methods / MINFLUX data analysis, page 5):

“To estimate the localization precision of a measurement as the third quantification parameter, the standard deviation σ_{rr} was calculated for each molecule (at least 5 localizations with the same exported parameter TID) as $\sigma_{rr} = \sqrt{\sigma_x^2 + \sigma_y^2}$ with the standard deviations of the xx- and yy- coordinates as determined by the microscope (exported parameter POS). The median σ_{rr} represents the stated localization precision. The combined localization precision was estimated as $\sigma_{rrr} = \langle \sigma_{rr} \rangle / \sqrt{nn}$, i.e. the weighted average of the average single localization precision σ_{rr} divided by \sqrt{nn} and weighted by the occurrence of nn being the number of single localizations with the same TID.”

8. In the supplement line 301, 304, 425: The relative laser power of 14% seems strange to include, as it is specific to the device. It would be better to stick to absolute measurements and include an estimate of power density at the confocal spot. This can be measured with a power meter from Thorlabs.

We thank the reviewer for this suggestion. In the revised version of the manuscript, all laser powers are reported as μ W deposited in the sample.

Reviewer #2:

Remarks to the Author:

This well written manuscript by Ostersehl et al describes a combination of MINFLUX, a next generation super-resolution fluorescence imaging method, with DNA-PAINT, a concept for single-molecule localization based super-resolution microscopy, building on transient binding of fluorescent molecules to the target molecules to be imaged. The motivation of this combined concept, and the synergies which come with it, are convincingly and clearly described. The combined concept, DNA-PAINT MINFLUX Nanoscopy, is applied on several different cellular samples, where the specific advantages of the concept, such as its abilities for 3D imaging, imaging of densely packed molecules and multiplexing (by subsequently adding, and washing away, different orthogonal strands targeting different target docking strands) are clearly demonstrated. The concept thus represents an important new tool and a significant advance in the field of fluorescence imaging.

We thank the reviewer for the positive view on our manuscript.

To further evaluate the synergies, the authors then investigated how certain key parameters influence the performance, where the performance was assessed based on three variables: i) time between valid events ($t(\text{btw})$), ii) center-frequency-ratio (CFR), and iii) localization precision ($\sigma(r)$). This performance evaluation is important and highly relevant for all scientists who want to apply this concept in the future. However, the evaluation would be more useful if the outcome could be presented in somewhat more general and transparent measures. In the evaluation, presented mainly in the SI and supplementary notes, several trends in the graphs essentially reflect specific (but not mentioned) settings of the MINFLUX instrument software used (e.g. Figs SN1a, 1b, 2a, 2b and 3a). Also, for several of the parameters investigated, their optimal settings seem difficult to more generally translate into other experimental conditions.

We fully agree that it is a difficult balance between a more general description of the evaluation of the MINFLUX parameters and a description of the specific settings tailored to the microscope used.

Because the microscope used is the only MINFLUX system available on the market and because it is a new and largely untested technology, we believe that it is beneficial for the readers to have information also on specific settings. In the revised manuscript, all settings are detailed in the full MINFLUX imaging sequence given in Supplementary data set 1. Key parameters of the MINFLUX sequence are now pointed out in the paragraph *Supplementary Methods/MINFLUX sequences* (page 7). Practically, information on these settings may help to set up experiments and therefore we prefer to keep information on these specific settings in the Supplemental Information.

In order to provide an additional more general parameter that can be used to determine the performance of a MINFLUX microscope, we report in the revised manuscript additionally on the measured background fluorescence. For the revision, we added three new panels to the Supplementary Notes (Suppl. Note Fig. I d, Suppl. Note Fig. II d, Suppl. Note Fig. IV d), that report on the influence of the pinhole size, the laser power and the imager concentration on the background (f_{bg}).

In addition, we added a paragraph to the Supplementary Information which puts this systematic analysis into a more general context (pages 19-20).

In order to allow readers to analyze the data themselves and to be as transparent as possible, we not only included the entire MINFLUX sequence to the Supplementary Information, but also uploaded the entire analysis software-suite including all localization data (<https://doi.org/10.5281/zenodo.6396988>).

The laser powers should preferably be directly stated in their units in the graphs, not percentages, and it would also be useful to know what excitation intensities they correspond to in the sample.

We thank the reviewer for this suggestion. In the revised version of the manuscript, all laser powers are reported as μW deposited in the sample.

A good imager concentration is concluded to be around 2nM. How much will this concentration depend on the dissociation constant (KD) of the imager strand to the docking strand of the target, and what are the dissociation constants for the different strands used? How would different KDs affect the optimal setting of the other parameters studied, and to what extent will it also be a parameter to consider in the choice of imager concentration, in addition to target binding site density?

Yes, absolutely, the concentration will depend on the KD of the imager strand to the docking strand. Unfortunately, we do not know the KD, as the manufacturer of these strands (Massive Photonics, Graefeling, Germany) does not provide information on their sequence or their KD.

The effects of different KDs on the imaging parameters in DNA-PAINT nanoscopy have been investigated previously. Many of these findings can be translated to DNA-PAINT MINFLUX nanoscopy. To address the issues raised by the reviewer, we added an entire new paragraph (*Possible further improvements*) to the end of the Supplementary Notes (see page 20), providing additional references and discussing strategies to modify the imaging parameters by using other imager and docking strands as well as further modifications in using PAINT.

In conclusion, this manuscript presents an elegant and useful concept, a significant advance in fluorescence-based cellular imaging. With some clarifications and added information on how the key parameters influence its performance, this manuscript will likely be of large interest and value to scientists in the field of cellular imaging.

Thank you for your helpful and supportive comments.

Minor points:

- P.5, lines 127-128: change “a single binding event” to “single binding events”?

Done.

- SI, p.10, line 298: four fold?

Indeed, the text was misleading. The increase is in fact six-fold, because at the last iteration of the sequence, the power is six times higher than in the first iteration.

We clarified this in the text.

This is now stated in the legend to Supplementary Note Figure I.

In addition, we now explain the power increase with each iteration in a new table in the paragraph *Supplementary Methods/MINFLUX sequences* (page 7).

Reviewer #3:

Remarks to the Author:

In this brief communication, two existing approaches--DNA-PAINT & MINFLUX--are integrated to improve the latter. Conventional MINFLUX is limited to imaging two fluorescence channels, but by adopting a DNA hybridization scheme with sequential imaging cycles, this limitation is overcome. The authors demonstrate 3D imaging of three proteins in fixed human cells, although theoretically the number of species that can be imaged is unlimited. The manuscript is well written and fits the scope and readership of Nature Methods, but a more convincing visual and quantitative comparison among MINFLUX, DNA-PAINT, and DNA-PAINT MINFLUX should be included.

We thank this reviewer for the positive view on our manuscript and the helpful suggestions. We provide an extensive comparison of DNA-PAINT nanoscopy, conventional (dSTORM) MINFLUX nanoscopy, and DNA-PAINT MINFLUX nanoscopy, as detailed in the answer below.

Major comments

1. Figure 1 compares (diffraction limited) confocal imaging with DNA-PAINT MINFLUX and the latter performs better. However, as both DNA-PAINT & MINFLUX individually also outperform confocal imaging, this result was to be expected. To understand what the impact of combining DNA-PAINT with MINFLUX is, a visual comparison between all three--DNA-PAINT only, MINFLUX only, and DNA-PAINT MINFLUX--should be provided. For example, does the integration of MINFLUX with DNA-PAINT lower the resolution due to the linkage error induced by the DNA docking strand? Do they collect fewer localizations, because the total acquisition time is longer?

We thank the reviewer for this suggestion. To address this point we have added an entire new Figure (Supplementary Fig. 1). It provides a detailed theoretical comparison of DNA-PAINT nanoscopy, conventional (dSTORM) MINFLUX nanoscopy, and DNA-PAINT MINFLUX nanoscopy. The figure details differences between the methods with regard to several performance parameters. In the revised main manuscript text we now detail the synergistic impact of combining DNA-PAINT with MINFLUX recordings (line 56, page 2).

To experimentally compare standard Alexa Fluor 647 based MINFLUX nanoscopy with the new DNA-PAINT MINFLUX nanoscopy, we performed additional experiments. Fig. 1d and 1e show a comparison of dSTORM MINFLUX and DNA-PAINT MINFLUX recordings of the same cellular structure (Nup96-GFP). The image quality was the same and similar localization precisions were achieved (Supplementary Table 1).

Parameters such as the number of localizations, bleaching, and acquisition times depend on numerous experimental factors such as the buffer conditions, fluorophores, target densities, and light intensities (all of which are different for dSTORM MINFLUX and DNA-PAINT MINFLUX). This makes a proper comparison multidimensional and very complex. As the main message of this Brief Communication is the first demonstration of DNA-PAINT MINFLUX nanoscopy, we believe that such a detailed comparison would be out of the scope of this manuscript.

2. Performance metrics, such as the resolution, are only reported for DNA-PAINT MINFLUX. For potential future users of DNA-PAINT MINFLUX to make an informed decision on what method would be best for them and showcase how DNA-PAINT MINFLUX exploits a synergistic effect, the authors should include a table/figure with quantitative comparison of DNA-PAINT, MINFLUX, and DNA-PAINT MINFLUX. Metrics such as, number of species/colours that can be imaged, resolution/localization precision, acquisition time, etc. can be included.

In the revised manuscript, the new Supplementary Fig. 1 explicitly mentions metrics such as number of colors, attainable localization precision, but also limitations such as the requirement for specific buffer conditions or the need for specific illumination schemes. We believe that this matrix supports an informed decision on the choice of a suitable method to experimentally address a specific question.

We have also added a short paragraph to the main text describing the synergies achieved by combining DNA-PAINT and MINFLUX.

It reads (Page 2):

“We reasoned that by combining DNA-PAINT with MINFLUX recording, we could synergistically benefit from the advantages of both methods. As in the current MINFLUX nanoscopy implementations, the ‘background’ fluorescence stemming from diffusing imager strands is suppressed by the confocal pinhole, DNA-PAINT MINFLUX nanoscopy can be used in the far-field mode. DNA-PAINT MINFLUX nanoscopy is expected to provide the same single-digit nanometer resolution as conventional MINFLUX nanoscopy. Because the state-switching kinetics are determined by the binding of an imager strand to a docking strand, no dedicated buffer systems are required, and the kinetics can be adapted to the density of the targets by tuning the imager concentration. As in conventional MINFLUX nanoscopy using photoswitchable dyes, also in DNA-PAINT MINFLUX nanoscopy the individual localizations are recorded one-by-one. Thus the imaging time scales with the number of targets, making single-beam scanning MINFLUX particularly suited for recording small regions of interest. Another intrinsic benefit of using PAINT is the fact that when densely packed molecules are imaged, successive fluorophore docking avoids the interaction of fluorophores belonging to neighboring target molecules. Hence co-activation and mutual fluorophore quenching is largely avoided. Finally, as multiple orthogonal imager strands can be applied sequentially, each binding to a different docking strand (Exchange DNA-PAINT)¹⁷, addressing multiple targets should also be straightforward. For an overview of synergies, see also Supplementary Fig. 1.”

Of course, it would be possible to go more into detail, but we believe that a more detailed comparison would be better suited for a future review than for a Brief Communication.

3. Throughout the study, a very low imager strand concentration of 0.5 - 2.5 nM is used, whereas most DNA-PAINT studies use around 10 nM. Even with 10 nM, the required acquisition time can already be on the order of hours, and this lengthy acquisition time is a major limitation of DNA-PAINT. The authors here require imaging times of up to 7 hours (P4L113).

The reviewer is right, the current implementation of MINFLUX is inherently slow for larger fields of view. This is clearly stated on several occasions throughout the

manuscript. By modifying the MINFLUX sequence, but also by adapting the labeling strategy, there are options to speed up the imaging process within certain limits. This is discussed in a new paragraph at the end of the Supplementary Notes.

3.1. Could the authors elaborate on what implications this has for the potential of DNA-PAINT MINFLUX and what applications are currently within reach (and which are not)?

Thank you for raising this point. MINFLUX, and DNA-PAINT MINFLUX is particularly suited for small ROIs, rather than for whole cells. This is now stated in the main manuscript.

It reads (line 64, page 3):

“As in conventional MINFLUX nanoscopy using photoswitchable dyes, also in DNA-PAINT MINFLUX nanoscopy the individual localizations are recorded one-by-one. Thus the imaging time scales with the number of targets, making single-beam scanning MINFLUX particularly suited for recording small regions of interest.”

3.2. Several strategies to reduce the acquisition time have been developed in recent years, such as optimising sequence design, buffer composition, imager strand concentration or used protein-assisted strand preforming. Would the authors briefly discuss which of these strategies might be included in later iterations of DNA-PAINT MINFLUX?

Thank you for this suggestion. For the revised version of the manuscript, we added a new paragraph to the Supplemental Notes (*Possible further improvements*) that discusses options to speed up the imaging process. This includes other PAINT variants (Förster resonance energy transfer (FRET)-based probes, caged, photo-activatable dyes, fluorogenic DNA-PAINT probes, preloading of DNA-PAINT imager strands with Argonaute proteins, and improved sequences), modifications in the MINFLUX recording sequence, and ultimately parallelization.

4. After the introduction, the first thing mentioned is: “we first explored the influence of a number of key parameters, such as laser power, confocal pinhole size and imager concentration on MINFLUX imaging with DNA-PAINT. Specifically, we determined the influence of these variables on i) the time btw between valid events, ii) the center-frequency-ratio (CFR), a filter parameter for localizations during image acquisition⁴, and iii) the localization precision σ .” However, later the analysis of these parameters is reported in the Supplementary and in the main only the final recommended values are provided. If the authors want to place such an emphasis on these parameters, this referee suggests to include a more detailed analysis & substantiation in the main text and mention this parameter analysis in the abstract. Furthermore, in line with an earlier comment, this referee suggests to put the found values for laser power, pinhole size, and imager concentration into context by providing comparative values for DNA-PAINT and/or conventional MINFLUX. If this makes the length of this article not fit Brief Communications, the authors may either not emphasise these parameters or consider submitting revision in the form of Research Article.

We do see the point raised by the reviewer. We re-wrote the introduction to reduce the emphasis on the analysis of the parameters. The main manuscript is shorter and more legible due to this modification.

We believe that the reported data fit best to the Brief Communications format, and rather would prefer not to inflate the manuscript to a full Research Article. Therefore, we prefer to follow the suggestion of the reviewer not to emphasize these parameters in order to keep the paper in a short and compact format.

Minor comments

1. As Nature Methods wishes its publications to contain a technical description that is adequate for reproduction. Would the authors make code & data directly accessible online (e.g. github) instead of upon request?

Yes, all code and all data will be made available via zenodo.org. Concretely, a comprehensive software package (written in Matlab) for drift correction, precision estimation as well as CFR and FRC calculations will be made accessible online. The software package also includes localization data for all figures presented in the manuscript.

The localization data and all custom codes used for image analysis are available at <https://doi.org/10.5281/zenodo.6396988>.

2. P2L43: typo in “a transient binding”, “a” should be removed.

Done.

Decision Letter, second revision: B

29th Apr 2022

Dear Stefan,

Your Brief Communication, "DNA-PAINT MINFLUX Nanoscopy", has now been seen again by the three reviewers. As you will see, two reviewers now approve publication of your manuscript, while referee 1 still has some additional information they would like to see added to the final version.

We therefore invite you to revise your manuscript to address these concerns before we make our final decision.

We hope to receive your revised paper within six weeks. If you cannot send it within this time, please let us know. In this event, we will still be happy to reconsider your paper at a later date so long as nothing similar has been accepted for publication at Nature Methods or published elsewhere.

OPEN SCIENCE REQUIREMENTS

REPORTING SUMMARY AND EDITORIAL POLICY CHECKLISTS

DATA AVAILABILITY

All novel DNA and RNA sequencing data, protein sequences, genetic polymorphisms, linked genotype and phenotype data, gene expression data, macromolecular structures, and proteomics data must be deposited in a publicly accessible database, and accession codes and associated hyperlinks must be provided in the "Data Availability" section.

Please include a "Data availability" subsection in the Online Methods. This section should inform readers about the availability of the data used to support the conclusions of your study, including accession codes to public repositories, references to source data that may be published alongside the paper, unique identifiers such as URLs to data repository entries, or data set DOIs, and any other statement about data availability. At a minimum, you should include the following statement: "The data that support the findings of this study are available from the corresponding author upon request", describing which data is available upon request and mentioning any restrictions on availability. If DOIs are provided, please include these in the Reference list (authors, title, publisher (repository name), identifier, year). For more guidance on how to write this section please see:

<http://www.nature.com/authors/policies/data/data-availability-statements-data-citations.pdf>

CODE AVAILABILITY

Please include a "Code Availability" subsection in the Online Methods which details how your custom code is made available. Only in rare cases (where code is not central to the main conclusions of the paper) is the statement "available upon request" allowed (and reasons should be specified).

For more information on our code sharing policy and requirements, please see:
<https://www.nature.com/nature-research/editorial-policies/reporting-standards#availability-of-computer-code>

MATERIALS AVAILABILITY

ORCID

Sincerely,
Rita

Rita Strack, Ph.D.
Senior Editor
Nature Methods

Reviewers' Comments:

Reviewer #1:

Remarks to the Author:

The authors have addressed most of my concerns. However, we still disagree on some essential aspects.

I have previously encouraged the authors to include more quantitative data over just the reconstructions in Figures 1 and 2. I would strongly suggest: i) adding to these figures histograms that visually show what the localization precision is in x,y, and z; ii) adding a quantitative benchmark of the structures as the structures are known (e.g. do Figure 2 d,e contain significant artifacts?); iii) add an experiment with DNA-PAINT on DNA origami, so that the authors can assess if the localizations are unbiased in x,y,z i.e. are artifact-free.

Finally, I would suggest that the authors include the FRC table from the rebuttal in the supplement and make all the raw experimental data available online (i.e. not only the localizations).

Reviewer #2:

Remarks to the Author:

In this revised manuscript, the authors have in my view satisfactorily addressed the comments. It is recognized that it is a difficult balance between giving a generally translatable description and providing the specific optimal settings of the particular microscope used. The added paragraphs in the SI on p. 19-20 give more generally translated information as asked for, with more extensive information to be found in the Zenodo data bank. In the latter paragraph, the influence of the off- and on-rates of the DNA-PAINT probes is also reasonably clarified.

Thereby, with these clarifications and added information, this manuscript will likely be of large interest and use to the nanoscopy community, and it can be recommended for publication.

Reviewer #3:

Remarks to the Author:

The main suggestion of this referee was to include a more convincing visual and quantitative comparison among MINFLUX, DNA-PAINT, and DNA-PAINT MINFLUX. This has been sufficiently addressed through Fig. 1de, SI Fig. 1 and the added paragraph on page 2; and these additions have improved the manuscript. In addition, the impact of lengthy acquisition time for possible applications for DNA-PAINT MINFLUX is transparently stated and possible solutions are now included in the supplemental notes. Lastly, the section after the introduction has been reorganized and shortened, which this referee feels improves the focus and clarity of the manuscript.

Author Rebuttal, second revision: C

Point-by-point response

Reviewers' Comments:

Reviewer #1:

Remarks to the Author:

The authors have addressed most of my concerns. However, we still disagree on some essential aspects.

We are pleased that we have been able to address the reviewer's previous concerns and hope that we can also address the remaining points.

I have previously encouraged the authors to include more quantitative data over just the reconstructions in Figures 1 and 2. I would strongly suggest: i) adding to these figures histograms that visually show what the localization precision is in x,y, and z;

We agree with the reviewer that the histograms of the localization precisions of all data shown in the figures are useful additional information. To this end, we now added, additionally to the histograms for Figure 1 also the histograms for Figure 2. However, we do not think that it would benefit the main manuscript to have an additional, large and rather unimportant figure (all localization precisions are clearly stated in the main text and the more detailed localization precision information in the supplement is clearly referred to). Therefore, in the revised version of the manuscript the localization precision histograms remain in the Supplement as Supplementary Fig. 2. We leave it to the editorial board to decide whether this figure should be included in the main text.

ii) adding a quantitative benchmark of the structures as the structures are known (e.g. do Figure 2 d,e contain significant artifacts?);

It has been shown previously that the MINFLUX localization process does not produce spatially biased localizations (Balzarotti et al., 2017, Science; Gwosch et al, 2020, Nat Methods). As the MINFLUX localization process is independent of the labeling scheme, there is no reason to assume that the DNA-PAINT labeling method will introduce localization artifacts. Of course, and this has been extensively discussed in the literature, the labeling is likely to be imperfect. As this manuscript is about the introduction of DNA-PAINT MINFLUX nanoscopy, rather than about a detailed description of a specific cellular structure, we think that it is out of the scope of this study to examine the labeling efficiency and we would like to leave such a detailed analysis for a future study. However, we did add these points to the discussion section of the manuscript.

It now reads (page 5):

“Since fluorescence microscopes render nothing but the fluorophores in the sample, the concept of spatial resolution can only be applied to the fluorophores. To be able to draw meaningful conclusions about the target molecules at the <5 nm scale, the size and mobility of the linker between the molecule and the fluorophore have to be taken into account. To fully harness the nanometer optical resolution potential of MINFLUX nanoscopy, these sample parameters deserve further attention and improvement. In addition to the size of the label, in particular the completeness of the labelling and the

fraction of fluorophores that can be successfully localized must also be taken into account. DNA-PAINT MINFLUX makes it possible to localize each binding site several times. Therefore, missing localizations due to premature bleaching of the fluorophore are avoided with this technique.”

iii) add an experiment with DNA-PAINT on DNA origami, so that the authors can assess if the localizations are unbiased in x,y,z i.e. are artifact-free.

The MINFLUX localization process is independent of the labeling scheme. It has been shown previously (Balzarotti et al., 2017, Science; Gwosch et al, 2020, Nat Methods) that MINFLUX can be used to faithfully record DNA origamis. These studies showed that the MINFLUX localizations are unbiased in x,y,z, i.e. are artifact-free. Therefore, we believe that following this new request would not benefit the manuscript, since it will not bring any new information. To clarify the fact that the localization process in DNA-PAINT MINFLUX is the same as in previous MINFLUX implementations and gives unbiased localizations we now added to the main text (page 6):

“The localization process remains unchanged compared to previous implementations. Therefore, DNA-PAINT MINFLUX nanoscopy provides the same unbiased, high precision localization demonstrated in previous studies^{2,3}.”

Finally, I would suggest that the authors include the FRC table from the rebuttal in the supplement

As discussed in our earlier response to the suggestion to include FRC data, we do not believe that it is beneficial for the reader to show the FRC values in the Supplement as these values are highly structure-dependent. However, to comply with the reviewer's request, we have included all FRC values shown in the rebuttal FRC table in Supplementary Table 1.

and make all the raw experimental data available online (i.e. not only the localizations).

We added the raw experimental data to the online data at <https://doi.org/10.5281/zenodo.6562764>.

Reviewer #2:

Remarks to the Author:

In this revised manuscript, the authors have in my view satisfactorily addressed the comments. It is recognized that it is a difficult balance between giving a generally translatable description and providing the specific optimal settings of the particular microscope used. The added paragraphs in the SI on p. 19-20 give more generally translated information as asked for, with more extensive information to be found in the Zenodo data bank. In the latter paragraph, the influence of the off- and on-rates of the DNA-PAINT probes is also reasonably clarified. Thereby, with these clarifications and added information, this manuscript will likely be of large interest and use to the nanoscopy community, and it can be recommended for publication.

We are pleased that we could satisfactorily address the reviewer's comments and thank the reviewer for the positive view on our manuscript.

Reviewer #3:

Remarks to the Author:

The main suggestion of this referee was to include a more convincing visual and quantitative comparison among MINFLUX, DNA-PAINT, and DNA-PAINT MINFLUX. This has been sufficiently addressed through Fig. 1de, SI Fig. 1 and the added paragraph on page 2; and these additions have improved the manuscript. In addition, the impact of lengthy acquisition time for possible applications for DNA-PAINT MINFLUX is transparently stated and possible solutions are now included in the supplemental notes. Lastly, the section after the introduction has been reorganized and shortened, which this referee feels improves the focus and clarity of the manuscript.

We are pleased that we were able to respond satisfactorily to the reviewer's comments and thank the reviewer for the positive evaluation of our manuscript.

Decision Letter, third revision: C

23rd May 2022

Dear Stefan,

Thank you for submitting your revised manuscript "DNA-PAINT MINFLUX Nanoscopy" (NMETH-BC47719C). Based on your revisions, we'll be happy in principle to publish it in Nature Methods, pending minor revisions to comply with our editorial and formatting guidelines.

TRANSPARENT PEER REVIEW

Thank you again for your interest in Nature Methods Please do not hesitate to contact me if you have any questions.

Sincerely,
Rita

Rita Strack, Ph.D.
Senior Editor
Nature Methods

ORCID

8th Jun 2022

Decision Letter, fourth revision: D

15th Jul 2022

Dear Stefan,

I am pleased to inform you that your Brief Communication, "DNA-PAINT MINFLUX Nanoscopy", has now been accepted for publication in Nature Methods. Your paper is tentatively scheduled for publication in our September print issue, and will be published online prior to that. The received and accepted dates will be Nov 26, 2021 and July 15, 2022. This note is intended to let you know what to expect from us over the next month or so, and to let you know where to address any further questions.

Your paper will now be copyedited to ensure that it conforms to Nature Methods style. Once proofs are generated, they will be sent to you electronically and you will be asked to send a corrected version within 24 hours. It is extremely important that you let us know now whether you will be difficult to contact over the next month. If this is the case, we ask that you send us the contact information (email, phone and fax) of someone who will be able to check the proofs and deal with any last-minute problems.

If, when you receive your proof, you cannot meet the deadline, please inform us at rjsproduction@springernature.com immediately.

Once your manuscript is typeset and you have completed the appropriate grant of rights, you will receive a link to your electronic proof via email with a request to make any corrections within 48 hours. If, when you receive your proof, you cannot meet this deadline, please inform us at rjsproduction@springernature.com immediately.

Once your paper has been scheduled for online publication, the Nature press office will be in touch to confirm the details.

Content is published online weekly on Mondays and Thursdays, and the embargo is set at 16:00 London time (GMT)/11:00 am US Eastern time (EST) on the day of publication. If you need to know the exact publication date or when the news embargo will be lifted, please contact our press office after you have submitted your proof corrections. Now is the time to inform your Public Relations or Press Office about your paper, as they might be interested in promoting its publication. This will allow them time to prepare an accurate and satisfactory press release. Include your manuscript tracking number NMETH-BC47719D and the name of the journal, which they will need when they contact our office.

About one week before your paper is published online, we shall be distributing a press release to news organizations worldwide, which may include details of your work. We are happy for your institution or funding agency to prepare its own press release, but it must mention the embargo date and Nature Methods. Our Press Office will contact you closer to the time of publication, but if you or your Press Office have any inquiries in the meantime, please contact press@nature.com.

If you are active on Twitter, please e-mail me your and your coauthors' Twitter handles so that we may tag you when the paper is published.

Please note that *Nature Methods* is a Transformative Journal (TJ). Authors may publish their research with us through the traditional subscription access route or make their paper immediately open access through payment of an article-processing charge (APC). Authors will not be required to make a final decision about access to their article until it has been accepted. [Find out more about Transformative Journals](https://www.springernature.com/gp/open-research/transformative-journals)

To assist our authors in disseminating their research to the broader community, our SharedIt initiative provides you with a unique shareable link that will allow anyone (with or without a subscription) to read the published article. Recipients of the link with a subscription will also be able to download and print the PDF. As soon as your article is published, you will receive an automated email with your shareable link.

Please note that you and your coauthors may order reprints and single copies of the issue containing your article through Springer Nature Limited's reprint website, which is located at <http://www.nature.com/reprints/author-reprints.html>. If there are any questions about reprints please send an email to author-reprints@nature.com and someone will assist you.

Please feel free to contact me if you have questions about any of these points (but please note I will be away until August 1st, email Dr. Allison Doerr a.doerr@us.nature.com with any immediate concerns).

Best regards,
Rita

Rita Strack, Ph.D.
Senior Editor
Nature Methods